# Bundle-specific associations between white matter microstructure and Aβ and tau pathology in preclinical Alzheimer's disease

Alexa Pichet Binette[1,2]*, Guillaume Theaud[3], François Rheault[4], Maggie Roy[3], D Louis Collins[5], Johannes Levin[6,7], Hiroshi Mori[8], Jae Hong Lee[9], Martin Rhys Farlow[10], Peter Schofield[11,12], Jasmeer P Chhatwal[13], Colin L Masters[14], Tammie Benzinger[15,16], John Morris[15,16], Randall Bateman[15,16], John CS Breitner[1,2], Judes Poirier[1,2], Julie Gonneaud[2,17], Maxime Descoteaux[3], Sylvia Villeneuve[1,2,5]*, DIAN Study Group, PREVENT-AD Research Group

[1]Department of Psychiatry, Faculty of Medicine, McGill University, Montreal, Canada; [2]Douglas Mental Health University Institute, Montreal, Canada; [3]Sherbrooke Connectivity Imaging Laboratory (SCIL), Université de Sherbrooke, Sherbrooke, Canada; [4]Electrical Engineering, Vanderbilt University, Nashville, United States; [5]McConnell Brain Imaging Centre, Montreal Neurological Institute, Montreal, Canada; [6]Department of Neurology, Ludwig-Maximilians-Universität München, Munich, Germany; [7]German Center for Neurodegenerative Diseases (DZNE), Munich, Germany; [8]Department of Clinical Neuroscience, Osaka City University Medical School, Osaka, Japan; [9]Department of Neurology, Asan Medical Center, University of Ulsan College of Medicine, Seoul, Republic of Korea; [10]Department of Neurology, Indiana University, Bloomington, United States; [11]Neuroscience Research Australia, Sydney, Australia; [12]School of Medical Sciences, UNSW Sydney, Sydney, Australia; [13]Harvard Medical School, Massachusetts General Hospital, Boston, United States; [14]The Florey Institute of Neuroscience and Mental Health, University of Melbourne, Parkville, Australia; [15]Knight Alzheimer Disease Research Center, Washington University School of Medicine, St. Louis, United States; [16]Department of Neurology, Washington University School of Medicine, St. Louis, United States; [17]Normandie Univ, UNICAEN, INSERM, U1237, Institut Blood and Brain @ Caen-Normandie, Cyceron, Caen, France

*For correspondence:
alexa.pichetbinette@mail.mcgill.ca (APB);
sylvia.villeneuve@mcgill.ca (SV)

**Abstract** Beta-amyloid (Aβ) and tau proteins, the pathological hallmarks of Alzheimer's disease (AD), are believed to spread through connected regions of the brain. Combining diffusion imaging and positron emission tomography, we investigated associations between white matter microstructure specifically in bundles connecting regions where Aβ or tau accumulates and pathology. We focused on free-water-corrected diffusion measures in the anterior cingulum, posterior cingulum, and uncinate fasciculus in cognitively normal older adults at risk of sporadic AD and presymptomatic mutation carriers of autosomal dominant AD. In Aβ-positive or tau-positive groups, lower tissue fractional anisotropy and higher mean diffusivity related to greater Aβ and tau burden in both cohorts. Associations were found in the posterior cingulum and uncinate fasciculus in preclinical sporadic AD, and in the anterior and posterior cingulum in presymptomatic mutation carriers. These results suggest that microstructural alterations accompany pathological accumulation as early as the preclinical stage of both sporadic and autosomal dominant AD.

## Introduction

The progression of Alzheimer's disease (AD) includes a long asymptomatic phase, during which accumulating pathology is accompanied by various brain changes (*Jack et al., 2013*; *Sperling et al., 2011*). Beta-amyloid (Aβ) and tau proteins, the pathological hallmarks of the disease (*Duyckaerts et al., 2009*), start to accumulate decades before signs of cognitive impairment (*Bateman et al., 2012*; *Jansen et al., 2015*). Positron emission tomography (PET) can image both proteins in vivo (*Johnson et al., 2016*; *Klunk et al., 2004*; *Schöll et al., 2016*), and thus help in identifying the earliest brain changes associated with such pathologies. Aβ- and tau-PET tracer accumulate in distinct patterns of deposition that follows canonical brain networks/organization. Aβ develops a widespread pattern of deposition that recapitulates a default mode network-like pattern, accumulating early in the frontal and parietal lobes (*Mattsson et al., 2019*; *Villeneuve et al., 2015*). Tau accumulates in a more localized pattern that can start in the locus coeruleus before being detectable by tau-PET scans, followed by the medial temporal lobe in the preclinical phase of the disease, and spreading to the lateral temporal lobe and the rest of the brain in late stages (*Braak and Braak, 1991*; *Braak et al., 2011*). A prominent view is that pathology accumulates in functionally and/or structurally connected regions (*Franzmeier et al., 2019*; *Seeley et al., 2009*; *Sepulcre et al., 2017*; *Vogel et al., 2020*). Many studies have highlighted associations between Aβ- and tau-PET and brain functional activity early in the course of the disease (*Berron et al., 2020*; *Jones et al., 2017*; *Mormino et al., 2011*; *Sepulcre et al., 2017*). However, relations between pathology and white matter (WM) microstructure, as assessed by diffusion magnetic resonance imaging (MRI), remain elusive in preclinical AD. While WM degeneration is clearly apparent in the late symptomatic stages, how WM microstructure is affected early on in the disease process is less clear (*Sachdev et al., 2013*). Whole-brain diffusion MRI tractograms can represent the brain's WM architecture, but these are difficult to reconstruct because of extensive crossing of WM fibers and the complexity of tracking algorithms (*Rheault et al., 2020*). Recent advances in modeling and available algorithms have facilitated robust extraction of WM bundles with automated methods, thereby allowing their more precise investigation. As well, more specific measures have become available for analysis of WM (*Dyrby et al., 2014*). In particular, free-water (FW)-corrected diffusion tensor measures may offer better estimates of WM microstructure, yielding tissue-based fractional anisotropy and diffusivities after removing the FW contribution to each voxel (*Pasternak et al., 2009*).

We investigated diffusion-based measures of WM microstructure in bundles that connect cortical regions vulnerable to Aβ and tau deposition. We hypothesized that such bundles would show lower fractional anisotropy and higher diffusivity with more pathology as proxy of WM degeneration. We sought to expand upon the few studies linking preclinical AD pathology and WM microstructure and focused on bundles (defined a priori) connecting brain regions targeted early by AD pathology, notably the cingulum bundle (*Jacobs et al., 2018*). The latter is a large association bundle under the cingulate gyri that connects anterior to posterior cingulate regions and curves further into the parahippocampal gyri of the temporal lobe. This bundle is typically affected in symptomatic AD dementia (*Bubb et al., 2018*; *Jacobs et al., 2018*; *Kantarci et al., 2017*; *Roy et al., 2020*; *Wen et al., 2019*), and given its location, could be preferentially affected by Aβ, particularly in its anterior segment. Also of interest is the uncinate fasciculus, reported to be affected at the stage of mild cognitive impairment (*Mito et al., 2018*; *Roy et al., 2020*). This bundle connects parts of the limbic system, such as the hippocampus and amygdala, with the orbitofrontal cortex (*Von Der Heide et al., 2013*), brain regions thought to be key regions for tau and Aβ propagation, respectively (*van der Kant et al., 2020*). Our objective was to investigate associations between the microstructure in those bundles of interest and AD pathology in two cohorts of cognitively normal individuals at risk of AD, older adults at increased risk of sporadic AD and presymptomatic mutation carriers of autosomal dominant AD (ADAD).

## Results

### Approach and participants

Using state-of-the-art methods in diffusion MRI modeling, tractography, and tractometry, we aimed to better understand the associations between WM microstructure of key bundles in preclinical AD and deposition of Aβ and tau as measured by PET. We reasoned that the preclinical stage of AD should be the ideal point at which to study these questions, given that this is a period during which AD pathology is spreading but overall brain structure and function remain largely preserved. We therefore studied the preclinical stage of both late-onset sporadic AD and ADAD. Sporadic AD is the most common form of dementia, is multifactorial, and occurs most often in late life. ADAD is the rarer form of AD, caused by fully penetrant genetic mutations in PSEN1, PSEN2, or APP, that leads to Aβ accumulation up to 20 years prior to symptom onset (*Bateman et al., 2012*) and to onset of cognitive symptoms often in the 40s and early 50s. ADAD is considered a 'purer' form of preclinical AD since most mutation carriers do not exhibit age-associated co-pathologies. We studied a subset of 126 asymptomatic individuals at high risk of sporadic AD from the PRe-symptomatic EValuation of Experimental or Novel Treatments for AD (PREVENT-AD) cohort (*Breitner et al., 2016*) and 81 ADAD presymptomatic mutation carriers from the Dominantly Inherited Alzheimer's Network (DIAN) cohort (*Morris et al., 2012*). PREVENT-AD enrolls cognitively normal older adults at risk of sporadic AD given their parental or multiple-sibling family history of the disease. At the time of study, participants were on average 67.3 years of age, predominantly female, and highly educated (*Table 1*). Based on a threshold established previously using global cortical Aβ burden (*McSweeney et al., 2020*), 19% of the participants were considered Aβ-positive. We also considered the same proportion of participants with the highest entorhinal tau uptake to be tau-positive. DIAN enrolls adults from families with ADAD. Our focus was on presymptomatic mutation carriers, but analyses were also conducted in mutation non-carriers in order to rule out false-positive associations that could be due to off-target binding properties of the PET tracer. Mutation carriers were on average 34.5 years of age, while non-carriers were slightly older. Both groups had more than 50% female and were highly educated. For the DIAN cohort, we had access to Aβ-PET only, with 43% of the mutation carriers and none of the non-carriers classified as Aβ-positive (*Su et al., 2013*).

**Table 1.** Demographics.

| | PREVENT-AD (n = 126) | DIAN mutation carriers (n = 81) | DIAN mutation non-carriers (n = 96) |
|---|---|---|---|
| Age (years) | 67.3 ± 4.8 (68–83) | 34.5 ± 9.9 (18–61) | 39.3 ± 11.7 (19–69) |
| Sex F:M (%F) | 94:32 (75%) | 42:39 (52%) | 56:40 (58%) |
| *APOE4* carriers (%) | 50 (40%) | 24 (30%) | 26 (27%) |
| Education (years) | 15.2 ± 3.3 (7–24) | 15.2 ± 3.0 (10–24) | 15.1 ± 2.7 (10–26) |
| Handedness (n, % right-handed) | 114 (90%) | 69 (85%) | 82 (85%) |
| Systolic blood pressure | 129.0 ± 13.8 (100–164) | 122.5 ± 10.2 (95–155) | 123.5 ± 17.1 (90–190) |
| Diastolic blood pressure | 74.0 ± 8.1 (60–96) | 75.2 ± 8.8 (55–104) | 77.1 ± 10.1 (60–110) |
| Global Aβ SUVR[*] | 1.3 ± 0.3 (1.0–2.8) | 1.6 ± 0.7 (0.8–3.7) | 1.0 ± 0.1 (0.9–1.3) |
| Aβ-positive (%) | 24 (19%) | 35 (43%) | 0 (0%) |
| Entorhinal tau SUVR | 1.1 ± 0.1 (0.7–1.6) | NA | NA |
| Mini-Mental State Examination | 28.8 ± 1.2 (24–30) | 29.0 ± 1.3 (24–30) | 29.2 ± 1.2 (25–30) |
| Estimated years to symptom onset[†] | −5.7 ± 7.6 (−20.8 to 16.8) | −13.6 ± 8.3 (−31.5 to 11.8) | −7.4 ± 12.5 (−28.8 to 21.4) |

Values represent Mean ± SD (range). Participants with at least one ε4 allele were considered APOE4 positive. The Mini-Mental State Evaluation was administered at the same time as PET.

[*] Note that NAV4694 was used in PREVENT-AD and PIB was used in DIAN.

[†] Estimated years to symptom onset was calculated as the parent's age at dementia onset minus the age of the participant; four missing values in PREVENT-AD.

Aβ: beta-amyloid; APOE: apolipoprotein E; SUVR: standardized uptake value ratio; PET: positron emission tomography.

## Methodology overview

We extracted FW-corrected diffusion tensor measures in bundles of interest. We reconstructed each individual's whole-brain tractogram using high angular resolution diffusion imaging and fiber orientation distribution functions (fODFs), and employed automated tools to isolate the anterior cingulum, the posterior cingulum, and the uncinate fasciculus (*Garyfallidis et al., 2018*; *Rheault, 2020*; *Wassermann et al., 2016*). Tractometry then generated bundle-specific quantification of five WM properties (*Cousineau et al., 2017*; *Rheault et al., 2017*). These were tissue fractional anisotropy (FA$_T$), mean diffusivity (MD$_T$), axial diffusivity (AD$_T$), and radial diffusivity (RD$_T$). In each, 'T' represents tissue in these FW-corrected diffusion tensor measures. We also report the FW index, which is thought to reflect a measure of neuroinflammation (*Pasternak et al., 2009*). To investigate the relationships with AD pathology, we focused on typical measures of Aβ- and tau-PET, which is a global cortical Aβ burden (in PREVENT-AD and DIAN) and entorhinal tau tracer uptake (in PREVENT-AD only).

We first evaluated the partial correlations between WM microstructure and pathology at the whole group level, controlling for age, sex, and bundle volume. We then repeated these analyses while restricting them to participants with (Aβ-positive or tau-positive) and without (Aβ-negative or tau-negative) pathology. The analysis conducted in Aβ-positive or tau-positive groups was especially important for the PREVENT-AD cohort since participants free from pathology might never develop AD. In complementary analyses, to investigate whether the associations would be independent from gray matter (GM) neurodegeneration, we further added GM volume in brain regions connected by our bundles of interest as covariates in the regression models. Lastly, we evaluated whether similar associations could be detected with typical diffusion tensor measures, that is, FA, MD, AD, and RD (not corrected for FW).

An overview of the processing steps is shown in *Figure 1* and can be summarized as follows: in three a priori WM bundles of interest extracted in the left and right hemisphere from each participant's tractogram, we evaluated associations between five related microstructure measures and AD pathology measured with PET (global cortical Aβ and entorhinal tau). We analyzed all five WM microstructure measures to detect whether a consistent pattern of associations across measures emerges rather than focusing on one given measure.

## Associations in the uncinate fasciculus and posterior cingulum in PREVENT-AD Aβ-positive and tau-positive groups

In PREVENT-AD, at the level of the whole group, there were no associations between global cortical Aβ or entorhinal tau burden with any of the WM microstructure measures across the three bundles of interest (*Figure 2—source data 1*, *Figure 3—source data 1*). Associations were detected only in the participants considered as Aβ-positive or tau-positive. In the Aβ-positive group, controlling for age, sex, and bundle volume, lower FA$_T$, higher MD$_T$, and higher RD$_T$ in the left posterior cingulum and in the uncinate fasciculus were related to greater cortical Aβ burden (*Figure 2*, *Figure 2—source data 1*). Similar associations were present in the right uncinate fasciculus at trend level (p=0.06 for FA$_T$, MD$_T$, and RD$_T$). Further, in the posterior cingulum, tau-positive participants displayed the same pattern of associations aforementioned; in this group, lower FA$_T$, higher MD$_T$, and higher RD$_T$ in the left posterior cingulum related to greater entorhinal tau-PET tracer binding (*Figure 3*, *Figure 3—source data 1*). Associations in the right posterior cingulum in tau-positive participants were trend level (p=0.06 for FA$_T$, MD$_T$, and RD$_T$).

In the anterior cingulum, there were no associations between WM measures and pathology in either Aβ-positive or tau-positive participants (*Figures 2D–3D*). No association was found in the Aβ- and tau-negative groups.

## Associations in anterior and posterior cingulum in DIAN mutation carriers

In DIAN mutation carriers, associations were found at the group level between global Aβ burden and WM microstructure in the anterior cingulum, following the same pattern of associations as in PREVENT-AD. As such, lower FA$_T$, higher MD$_T$, and higher RD$_T$ related to greater cortical Aβ across all mutation carriers (partial R = −0.27 for FA$_T$ and 0.28 for MD$_T$ and RD$_T$, p=0.02; *Figure 4—source data 1*), but associations were higher when restricted to the Aβ-positive participants (*Figure 4A–D*).

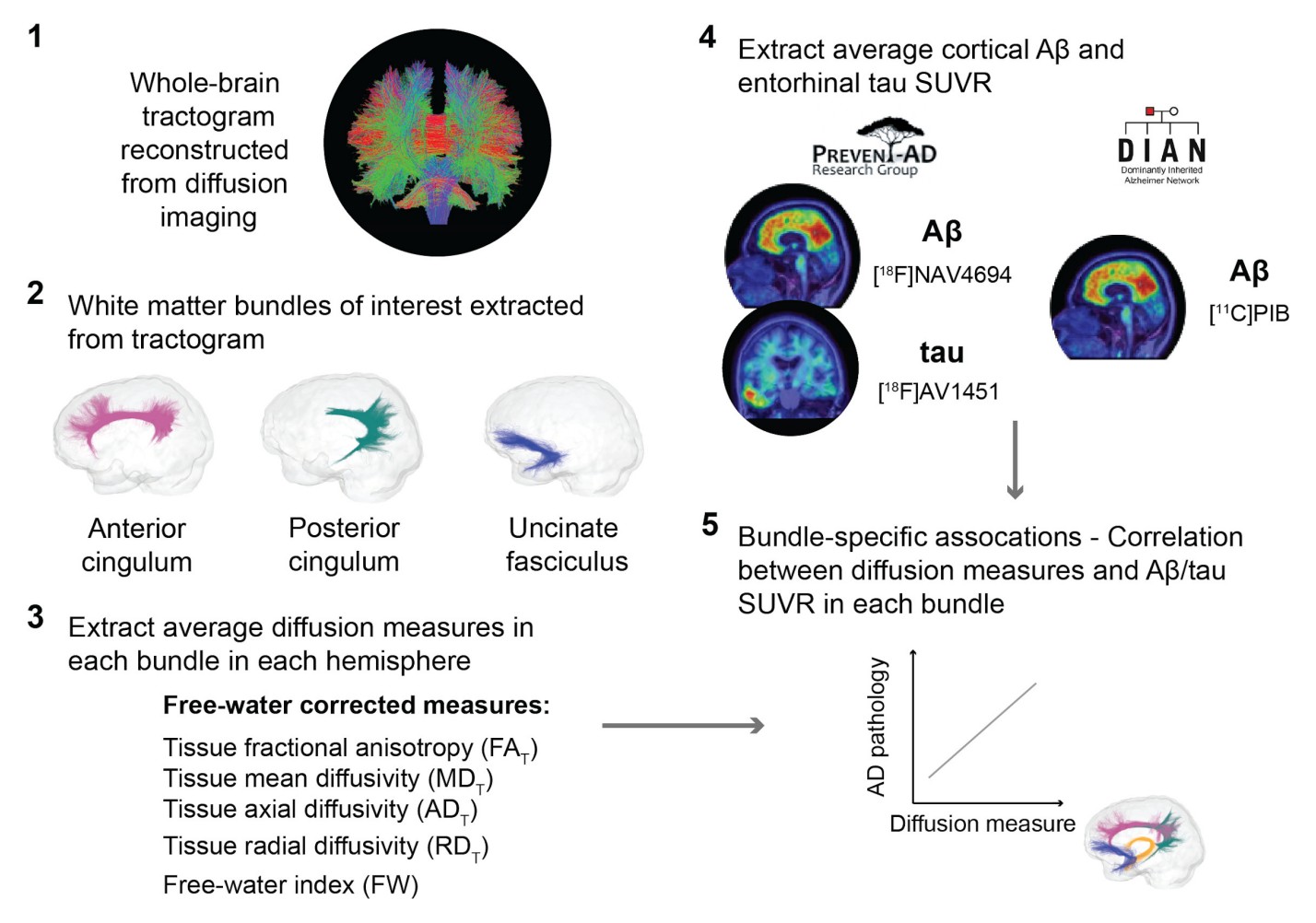

**Figure 1.** Overview of the processing steps. PREVENT-AD and DIAN participants were processed following the same pipeline. Whole-brain tractogram was reconstructed using the TractoFlow Atlas-Based Segmentation pipeline, and automated bundle extraction tools were used to extract the bundles of interest in each hemisphere. Free-water-corrected tensor measures were calculated for each bundle. Associations between white matter microstructure and global Aβ and entorhinal tau PET were then investigated. Aβ: beta-amyloid; PET: positron emission tomography; SUVR: standardized uptake value ratio.

Associations in Aβ-positive participants were also found in the right posterior cingulum (*Figure 4B–E*). Focusing only on the Aβ-negative group, microstructure measures in the posterior cingulum were associated with global cortical Aβ in the same directions as in the Aβ-positive group (partial R = −0.31 for $FA_T$ and 0.31 for $MD_T$ and $RD_T$, p=0.01). Of note, we did not find any associations between bundle microstructure and pathology in mutation non-carriers.

## Effect of GM atrophy on microstructure-pathology associations

We further wanted to evaluate whether significant associations between bundle microstructure and pathology in Aβ-positive or tau-positive participants were affected by GM atrophy. To do so, we added GM volume specifically in brain regions connected by the bundle of interest as an additional covariate. Partial correlations were thus controlled for age, sex, bundle volume, and GM volume. GM volume of the following regions were considered: the anterior and posterior cingulate cortex for the anterior cingulum, the precuneus and the parahippocampal gyrus for the posterior cingulum, and the medial orbitofrontal cortex and the parahippocampal gyrus for the uncinate fasciculus. In both PREVENT-AD and DIAN, further adjusting for GM volume did not change the significance of the microstructure measures in neither the anterior cingulum nor the uncinate fasciculus. The only

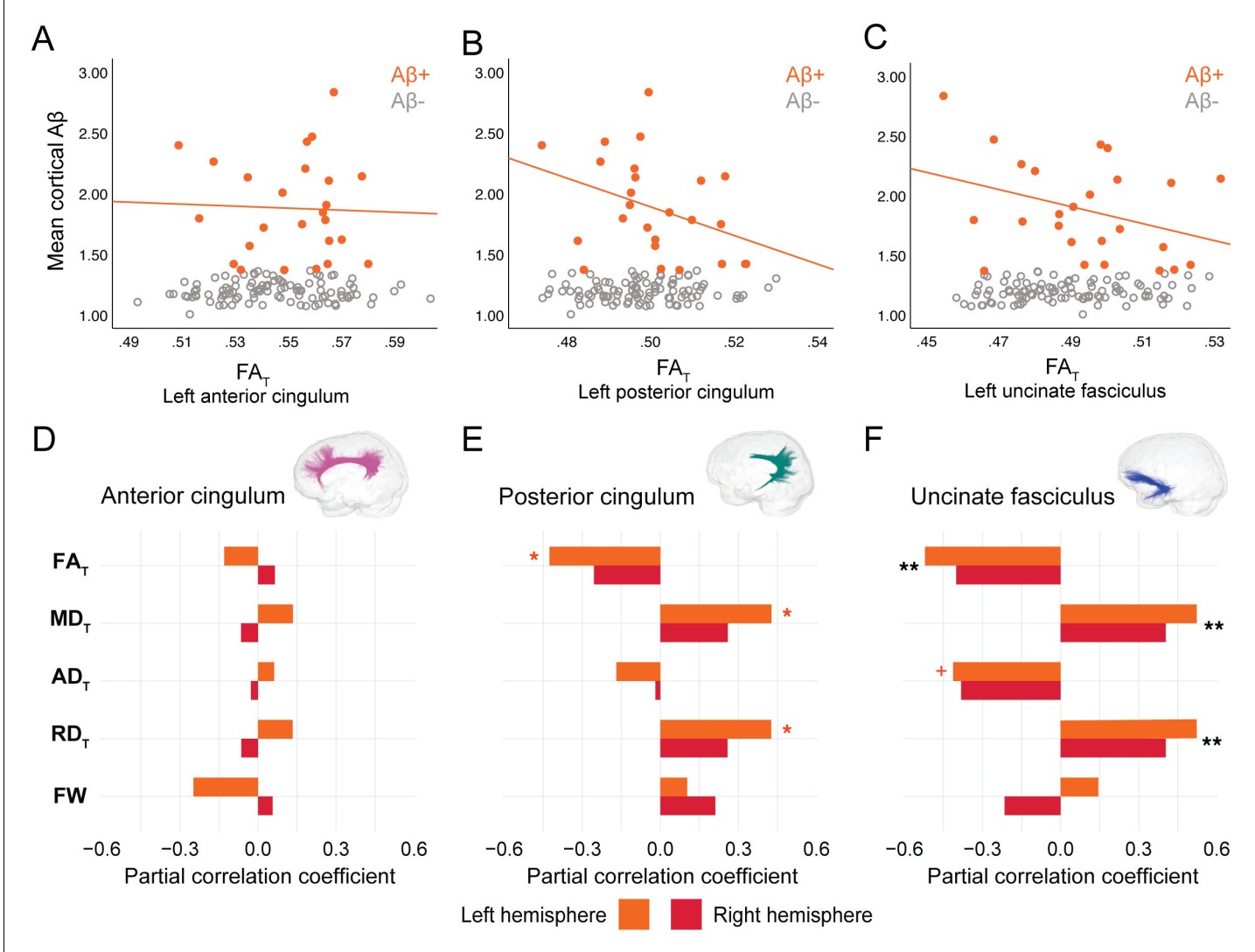

**Figure 2.** Associations between diffusion measures and Aβ burden in Aβ-positive PREVENT-AD participants. (**A–C**) Bivariate associations between $FA_T$ and global cortical Aβ in each bundle in the left hemisphere to show examples of raw values in PREVENT-AD. Data are represented for the full sample, with Aβ-positive in orange (our group of interest) and Aβ-negative in gray. (**D–F**) Partial correlations between diffusion measures (average diffusion measure in the bundle) and global cortical Aβ-PET controlling for age, sex, and bundle volume (divided by total intracranial volume) were performed in PREVENT Aβ-positive participants. Partial correlation coefficient for each diffusion measure in the right and left bundles is reported as bar graphs. Black asterisks highlight that associations are significant in both hemispheres, otherwise the color of the symbol matches the hemisphere where the association is significant. *p=0.05; ** 0.05 > p > 0.001; +p=0.06. See also *Figure 2—source data 1*. Aβ: beta-amyloid; $FA_T$: tissue fractional anisotropy; $MD_T$: tissue mean diffusivity; $AD_T$: tissue axial diffusivity; $RD_T$: tissue radial diffusivity; FW: free-water index; PET: positron emission tomography. The online version of this article includes the following source data for figure 2:

**Source data 1.** Associations between microstructure and beta-amyloid–positron emission tomography (Aβ-PET) in PREVENT-AD.

bundle where atrophy changed the original associations was the posterior cingulum, with GM volume of the precuneus as a covariate. In both PREVENT-AD and DIAN, in models including volume of the precuneus, microstructure properties were no longer related to pathology. When volume of the parahippocampal gyrus was a covariate in the models of the posterior cingulum, microstructure associations were unchanged compared to the initial analyses, with the exception of the ones with Aβ burden in PREVENT-AD (the contribution of diffusion measures became marginal, changing from p=0.05 to p=0.06). In complementary analyses, we evaluated whether GM volume related to Aβ- and tau-PET controlling for age and sex. The main significant associations were in the right precuneus or posterior cingulate in PREVENT-AD and in the right posterior cingulate in DIAN (*Table 2*).

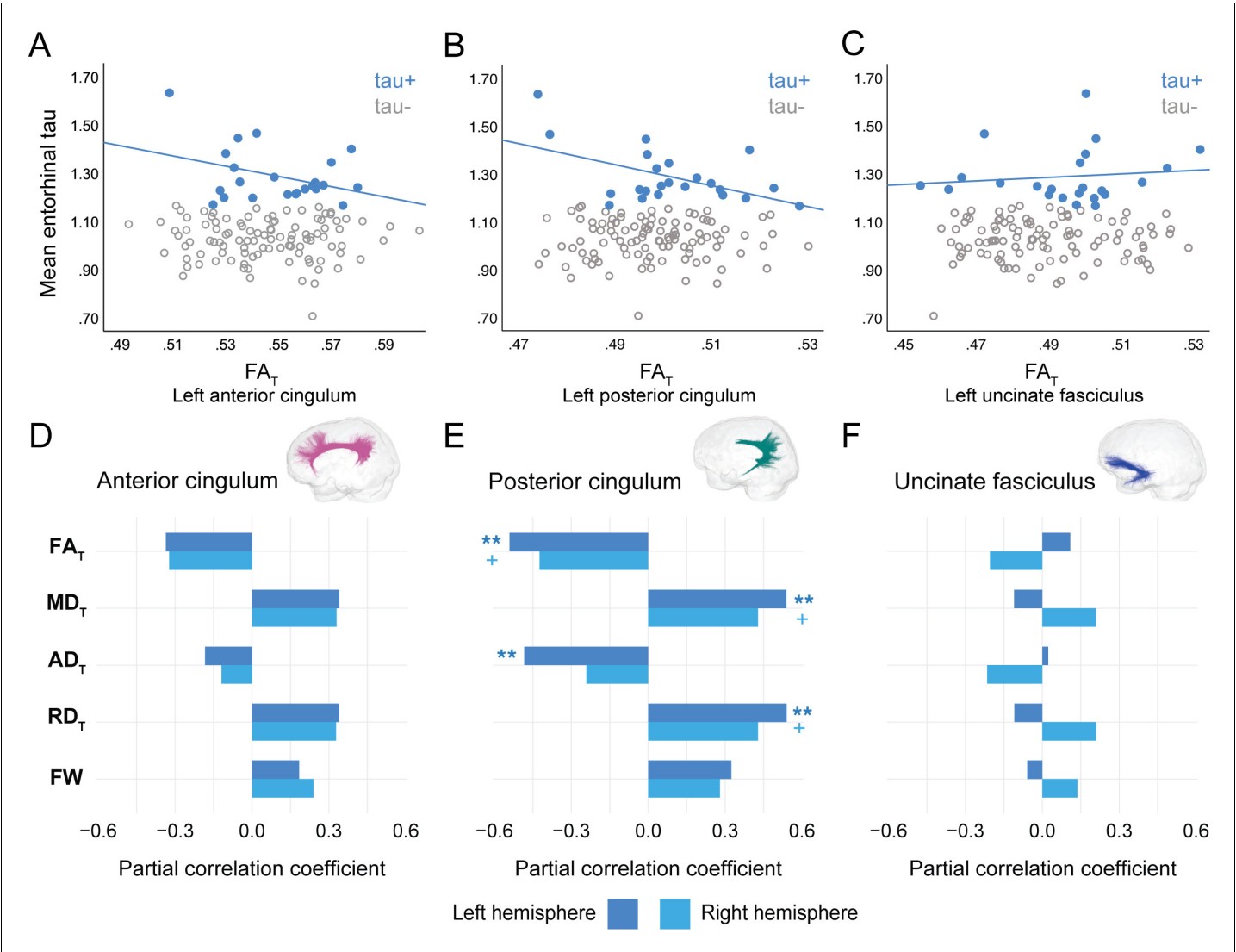

**Figure 3.** Associations between diffusion measures and entorhinal tau burden in tau-positive PREVENT-AD participants. (**A–C**) Bivariate associations between $FA_T$ and entorhinal tau in each bundle in the left hemisphere to show examples of raw values in PREVENT-AD. Data are represented for the full sample, with tau-positive in blue (our group of interest) and tau-negative in gray. (**D–F**) Partial correlations between diffusion measures (average diffusion measure in the bundle) and entorhinal tau-PET controlling for age, sex, and bundle volume (divided by total intracranial volume) were performed in PREVENT tau-positive participants. Partial correlation coefficient for each diffusion measure in the right and left bundles is reported as bar graphs. The color of the symbol on the bar graphs matches the hemisphere where the association is significant. *p=0.05; ** 0.05 > p > 0.001; +p=0.06. See also *Figure 3—source data 1*. $FA_T$: tissue fractional anisotropy; $MD_T$: tissue mean diffusivity; $AD_T$: tissue axial diffusivity; $RD_T$: tissue radial diffusivity; FW: free-water index; PET: positron emission tomography.

The online version of this article includes the following source data for figure 3:

**Source data 1.** Associations between microstructure and tau-positron emission tomography (tau-PET) in PREVENT-AD.

## Importance of advanced FW measures to these results

To evaluate the sensitivity of FW-corrected measures over the typical tensor measures, we tested whether similar associations with pathology exist with FA, MD, AD, and RD (i.e., not corrected for FW). Except for FA, which gave similar results to $FA_T$, MD, RD, and AD were not associated with pathology in any bundle (*Table 3*), suggesting that FW-corrected measures capture subtle WM microstructure alterations not always detectable with more classical diffusion tensor imaging (DTI) measures.

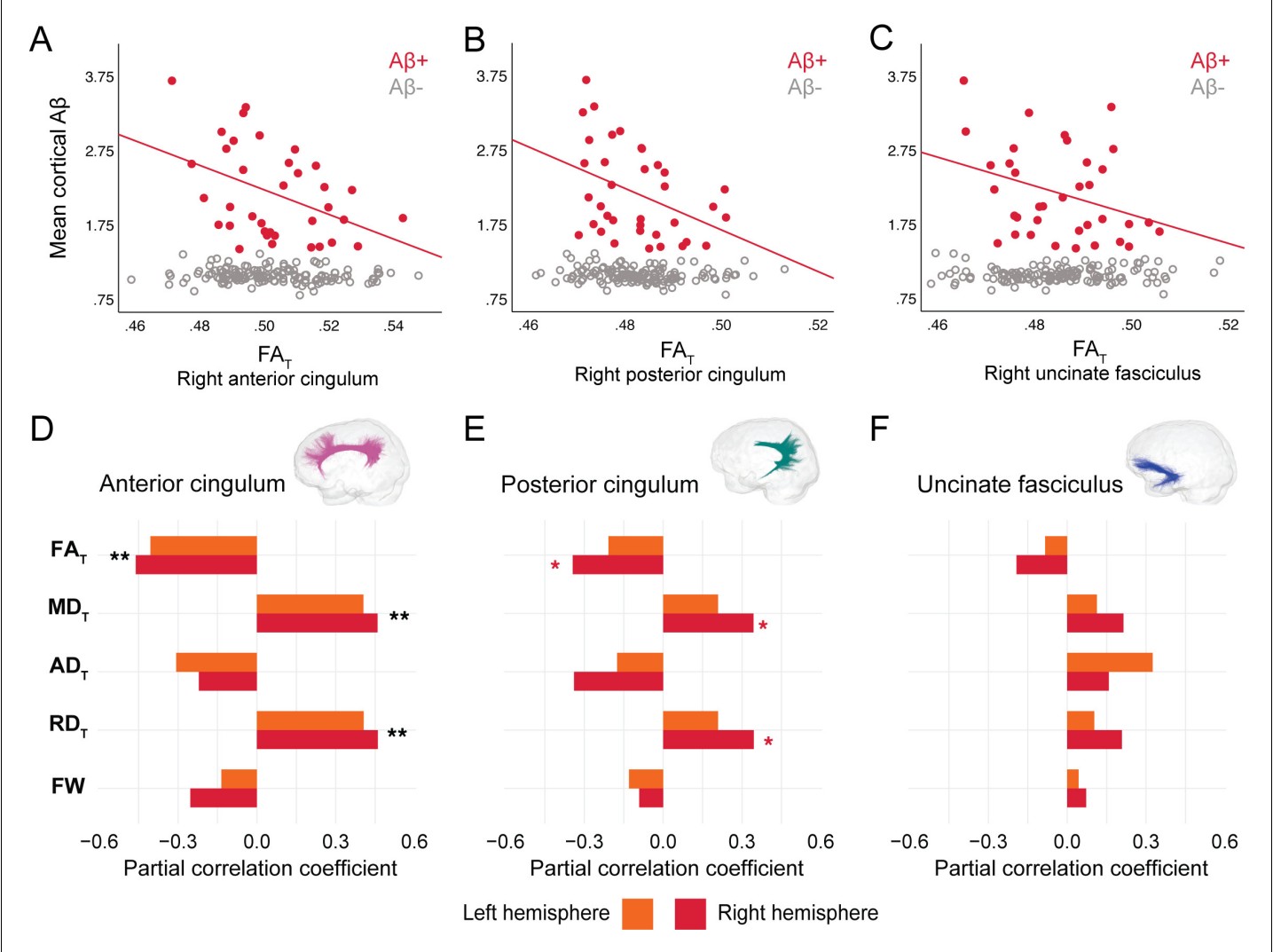

**Figure 4.** Associations between diffusion measures and Aβ burden in Aβ-positive DIAN mutation carriers. (A–C) Bivariate associations between $FA_T$ and global cortical Aβ in each bundle in the left hemisphere to show examples of raw values in DIAN. Data are represented for the full sample, with Aβ-positive in red (our group of interest) and Aβ-negative in gray. (D–F) Partial correlations between diffusion measures (average diffusion measure in the bundle) and global cortical Aβ-PET controlling for age, sex, and bundle volume (divided by total intracranial volume) were performed in DIAN Aβ-positive participants. Partial correlation coefficient for each diffusion measure in the right and left bundles is reported as bar graphs. Black asterisks highlight that associations are significant in both hemispheres, otherwise the color of the symbol matches the hemisphere where the association is significant *p=0.05; ** 0.05 > p > 0.001. See also *Figure 4—source data 1*. Aβ: beta-amyloid; $FA_T$: tissue fractional anisotropy; $MD_T$: tissue mean diffusivity; $AD_T$: tissue axial diffusivity; $RD_T$: tissue radial diffusivity; FW: free-water index; PET: positron emission tomography.

The online version of this article includes the following source data for figure 4:

**Source data 1.** Associations between microstructure and beta-amyloid–positron emission tomography (Aβ-PET) in DIAN.

## Discussion

The notion that AD pathology accumulates in connected regions in the brain has foundations in rodent models (*Ahmed et al., 2014*; *Jucker and Walker, 2018*; *Palop and Mucke, 2010*), and it is gaining credence in human neuroimaging studies. It is striking how pathology deposit in structurally or functionally connected regions (*Franzmeier et al., 2020*; *Seeley et al., 2009*; *Sepulcre et al., 2016*; *Vogel et al., 2020*). However, there is limited evidence on how WM microstructure in bundles linking those key pathology regions is affected in the early phases of AD. Combining Aβ- and tau-PET with recent advanced diffusion imaging analyses, we investigated WM microstructure in bundles (selected a priori) that connect key AD brain regions with Aβ and tau deposition. Our aim here was

**Table 2.** Associations between gray matter volume and Aβ- and tau-PET in PREVENT-AD and DIAN.

| | PREVENT-AD Aβ-positive | | PREVENT-AD Tau-positive | | DIAN Aβ-positive | |
|---|---|---|---|---|---|---|
| | $R_{partial}$ | p-value | $R_{partial}$ | p-value | $R_{partial}$ | p-value |
| *Left hemisphere* | | | | | | |
| Anterior cingulate | −0.239 | 0.271 | 0.032 | 0.891 | −0.207 | 0.248 |
| Posterior cingulate | −0.116 | 0.598 | −0.156 | 0.5 | −0.252 | 0.156 |
| Precuneus | −0.265 | 0.221 | **−0.439** | 0.047 | −0.307 | 0.082 |
| Parahippocampal gyrus | 0.114 | 0.605 | −0.073 | 0.753 | 0.079 | 0.661 |
| Medial orbitofrontal cortex | −0.468 | 0.024 | −0.197 | 0.392 | −0.174 | 0.334 |
| *Right hemisphere* | | | | | | |
| Anterior cingulate | −0.335 | 0.118 | −0.085 | 0.713 | −0.133 | 0.461 |
| Posterior cingulate | −0.342 | 0.111 | **−0.546** | 0.01 | **−0.352** | 0.045 |
| Precuneus | **−0.468** | 0.024 | **−0.491** | 0.024 | −0.287 | 0.105 |
| Parahippocampal gyrus | 0.014 | 0.948 | −0.213 | 0.355 | 0.16 | 0.373 |
| Medial orbitofrontal cortex | −0.358 | 0.093 | −0.237 | 0.301 | 0.014 | 0.94 |

$R_{partial}$ and p-values from regression models investigating associations between gray matter volume (divided by total intracranial volume; independent variable) and pathology (dependent variable) in Aβ-positive or tau-positive participants in PREVENT-AD and DIAN. Models included age and sex as covariates.

Aβ: beta-amyloid; PET: positron emission tomography.

not to test the spreading hypothesis per se, but, assuming that this hypothesis is correct, to focus on local effects of microstructure alterations and the presence of AD pathology in the preclinical stage of the disease. We investigated diffusion–PET associations in a cohort of asymptomatic older adults at risk of AD dementia due to their family history of sporadic AD and presymptomatic ADAD mutation carriers. In both cohorts, we found lower $FA_T$, higher $MD_T$, and higher $RD_T$ were related to greater pathology. In PREVENT-AD, associations were found in the uncinate fasciculus and the posterior cingulum, whereas in DIAN associations were found in the anterior and posterior segments of the cingulum. Furthermore, in the PREVENT-AD the associations were restricted to participants with significant AD pathology. These results suggest that significant levels of Aβ- and tau-PET tracer binding are associated with WM neurodegeneration both in the preclinical phase of sporadic AD and ADAD.

Our 'bundle-specific' approach through tractography and tractometry suggests topographical relationships between pathology and WM microstructural alterations in the early stage of AD and complements the typical approach of voxel-wise analyses (*Harrison et al., 2020*; *Zhang et al., 2019*). Using more precise tissue measures with FW corrected as opposed to classical diffusion tensor measures strengthened our findings, further highlighting the relevance of novel methods. Most of the models proposing a cascade of events over the course of AD have not included WM alterations (*Iturria-Medina et al., 2016*; *Jack et al., 2013*). One exception being a recent model in ADAD that included diffusivity, with higher MD being detectable 5–10 years prior to symptom onset (*Araque Caballero et al., 2018*). Although the current study design precludes us from staging when microstructure starts to change, our findings suggest that WM degeneration already occurs with early pathology accumulation prior to symptom onset both in the sporadic and the autosomal dominant forms of AD.

The observed associations follow the classical pattern of degeneration that is characterized by lower anisotropy and higher diffusivity, representing loss of coherence in the WM microstructure with AD progression (*Badea et al., 2016*; *Caso et al., 2016*; *Sexton et al., 2011*). This pattern of WM degeneration develops invariably along the AD spectrum (*Amlien and Fjell, 2014*; *Pereira et al., 2019*), with changes often becoming detectable only in the mild cognitive impairment and dementia stages (*Mito et al., 2018*; *Song et al., 2018*; *Wang et al., 2019*; *Wen et al., 2019*), and rarely in Aβ-positive cognitively normal participants (*Rieckmann et al., 2016*; *Vipin et al., 2019*). Our results further emphasize that in presymptomatic populations associations start to be

**Table 3.** Associations between typical tensor measures and Aβ- and tau-PET in PREVENT-AD and DIAN.

**PREVENT-AD**

**Aβ-positive**

| | Anterior cingulum | | Posterior cingulum | | Uncinate fasciculus | |
|---|---|---|---|---|---|---|
| | $R_{partial}$ | p-value | $R_{partial}$ | p-value | $R_{partial}$ | p-value |
| *Left hemisphere* | | | | | | |
| FA | −0.128 | 0.571 | −0.429 | 0.046 | −0.526 | 0.012 |
| MD | 0.022 | 0.923 | 0.18 | 0.424 | 0.32 | 0.147 |
| AD | −0.106 | 0.637 | −0.315 | 0.153 | −0.305 | 0.168 |
| RD | 0.081 | 0.721 | 0.305 | 0.168 | 0.448 | 0.037 |
| *Right hemisphere* | | | | | | |
| FA | −0.021 | 0.927 | −0.3 | 0.175 | −0.566 | 0.006 |
| MD | −0.048 | 0.831 | 0.131 | 0.56 | 0.078 | 0.729 |
| AD | −0.067 | 0.766 | −0.102 | 0.651 | −0.381 | 0.08 |
| RD | −0.019 | 0.931 | 0.221 | 0.322 | 0.344 | 0.116 |

**Tau-positive**

| | Anterior cingulum | | Posterior cingulum | | Uncinate fasciculus | |
|---|---|---|---|---|---|---|
| | $R_{partial}$ | p-value | $R_{partial}$ | p-value | $R_{partial}$ | p-value |
| *Left hemisphere* | | | | | | |
| FA | −0.465 | 0.039 | −0.523 | 0.018 | 0.108 | 0.65 |
| MD | 0.511 | 0.021 | 0.396 | 0.084 | 0.054 | 0.821 |
| AD | 0.112 | 0.639 | 0.082 | 0.731 | 0.206 | 0.384 |
| RD | 0.546 | 0.013 | 0.465 | 0.039 | −0.014 | 0.953 |
| *Right hemisphere* | | | | | | |
| FA | −0.461 | 0.041 | −0.487 | 0.029 | −0.399 | 0.081 |
| MD | 0.416 | 0.068 | 0.372 | 0.106 | 0.211 | 0.372 |
| AD | 0.012 | 0.959 | 0.157 | 0.508 | −0.011 | 0.965 |
| RD | 0.495 | 0.027 | 0.444 | 0.05 | 0.311 | 0.182 |

**DIAN**

**Aβ-positive**

| | Anterior cingulum | | Posterior cingulum | | Uncinate fasciculus | |
|---|---|---|---|---|---|---|
| | $R_{partial}$ | p-value | $R_{partial}$ | p-value | $R_{partial}$ | p-value |
| *Left hemisphere* | | | | | | |
| FA | −0.373 | 0.035 | −0.192 | 0.293 | −0.031 | 0.868 |
| MD | 0.15 | 0.413 | −0.051 | 0.783 | −0.015 | 0.935 |
| AD | −0.196 | 0.283 | −0.226 | 0.213 | −0.067 | 0.716 |
| RD | 0.318 | 0.076 | 0.049 | 0.79 | 0.021 | 0.91 |
| *Right hemisphere* | | | | | | |
| FA | −0.452 | 0.009 | −0.4 | 0.023 | −0.35 | 0.049 |
| MD | 0.122 | 0.505 | −0.002 | 0.991 | 0.108 | 0.558 |
| AD | −0.239 | 0.188 | −0.219 | 0.229 | −0.098 | 0.595 |
| RD | 0.335 | 0.061 | 0.146 | 0.426 | 0.209 | 0.251 |

$R_{partial}$ and p-values from regression models investigating associations between each tensor measure (average diffusion measure in the bundle; independent variable) and pathology (dependent variable) in PREVENT-AD Aβ-positive or tau-positive participants and DIAN Aβ-positive participants. Models included age, sex, bundle volume (divided by total intracranial volume) as covariates.

Aβ: beta-amyloid; FA: fractional anisotropy; MD: mean diffusivity; AD: axial diffusivity; RD: radial diffusivity; PET: positron emission tomography.

detectable in individuals with high amount of pathology. In effect, most of the microstructure-pathology associations were restricted to the Aβ-positive or tau-positive participants. We should note that in the asymptomatic stage there is also evidence of WM alterations opposing the typical degeneration pattern, suggesting a possible biphasic relationship over the course of the disease (*Fortea et al., 2010*; *Montal et al., 2018*; *Wearn et al., 2020*). For instance, hypertrophy, glial activation, neuronal or glial swelling have been attributed higher anisotropy and lower diffusivity in the asymptomatic phase (*Fortea et al., 2010*; *Montal et al., 2018*). The biomarker status (i.e., Aβ-positive or negative) might be important to disentangle such early processes (*Dong et al., 2020*; *Racine et al., 2014*). Not dichotomizing by pathology status might obscure some associations in the early disease stages, as shown here.

The bundle that was consistently affected in participants with high pathology in both cohorts was the posterior cingulum, a key bundle in AD (*Agosta et al., 2011*; *Caso et al., 2016*; *Zhuang et al., 2012*). The posterior cingulum is certainly altered in the symptomatic stage, and diffusivity in this bundle has also shown to be related to tau accumulation in preclinical individuals (*Jacobs et al., 2018*). In the PREVENT-AD cohort, the posterior segment of the bundle was the only region where tau-positive participants presented WM degeneration with greater entorhinal tau. In DIAN, although we did not have tau-PET, we hypothesize that the associations found in Aβ-positive in the posterior cingulum would be present with tau since mutation carriers harbor elevated tau binding in the precuneus (*Gordon et al., 2019*). In an attempt to explore whether associations were independent of atrophy in brain regions connected the bundles of interest, we also controlled for GM volume in such regions. In both cohorts, the precuneus is the only region where, when added as a covariate, microstructure was no longer related to pathology. Such finding might suggest that this critical region in AD pathophysiology might already be further along the degeneration process, with white and GM being affected. Our results both in preclinical sporadic and ADAD corroborate the idea that the precuneus/posterior cingulum, more largely part of the posterior default mode network or posterior-medial system, is a critical area in the cascading events of AD (*Berron et al., 2020*; *Jones et al., 2016*).

In DIAN, the other bundle where Aβ and WM measures were related was the anterior cingulum, another bundle connecting key regions where Aβ accumulates. In line with these results, similar associations have been found in DIAN using CSF Aβ (*Finsterwalder et al., 2020*). On the other hand, in PREVENT-AD, the strongest associations with Aβ were detected in the uncinate fasciculus. This bundle has an interesting anatomy, connecting regions at the intersection of both Aβ (frontal lobe) and tau (temporal lobe) deposition patterns in sporadic AD. We speculate that the particular localization of the uncinate fasciculus with regards to Aβ and tau deposition might confer early vulnerability to pathological insults. Further, the orbitofrontal cortex is not only a region where Aβ pathology accumulates early but is also a highly plastic late-developing region, typically affected in aging (*Fjell et al., 2014*; *Pichet Binette et al., 2020*). This might in part explain why the uncinate fasciculus is preferentially affected in preclinical sporadic AD compared to the younger mutation carriers of ADAD.

The direct investigation of WM fiber bundles and their microstructure was possible due to recent advances in diffusion imaging modeling, tractography, bundle extraction, and tractometry quantification. However, there are several limitations to these techniques and to our study. First, there are no common standards (yet) to extract predefined bundles from tractograms, and bundles with high curvature are more challenging to extract. To mitigate this challenge, we mostly relied on algorithms that use priors to help generate fuller bundles. We also used automated algorithms to increase reproducibility and performed rigorous visual inspection to make sure all algorithms yielded comparable bundles. The diffusion sequence, similar in both cohorts, relied on only one b-value, and future acquisitions with multiple b-values could further improve capturing fine-grained changes (*Pines et al., 2020*). We should also note that the PREVENT-AD cohort does not present highly elevated levels of tau, hence the deliberate choice of focusing on the proportion of participants with the highest levels rather than applying a definite cut-off. The sample size might not be huge, but we replicate all main findings in our two groups of interest. Both cohorts are also followed over time on

cognition and imaging, so future longitudinal studies can help clarify the sequence of events between pathology and WM changes in the preclinical and early symptomatic stages.

Overall, we used state-of-the-art analytical techniques to study associations between WM microstructure and Aβ- and tau-PET in key bundles affected in AD in the PREVENT-AD cohort of cognitively normal older adults whose strong family history of AD suggests an increased risk of subsequent dementia (*Cupples et al., 2004*; *Devi et al., 2000*) and in presymptomatic mutation carriers from the DIAN cohort. We highlighted the vulnerability of WM bundles to early presence of Aβ and tau proteins. More generally, the topography of our results aligns with the concept of retrogenesis, postulating that late-myelinated fibers, from temporal and neocortical regions, are affected first in the disease course and less resistant to neurodegeneration (*Alves et al., 2015*; *Bartzokis, 2004*; *Bartzokis, 2011*). As more studies highlight that WM changes might precede changes in GM (*Caso et al., 2016*; *Sachdev et al., 2013*), further investigations of WM microstructure in the early stages of AD will help understand better the complex pathogenesis of the disease.

## Materials and methods

### Participants
#### PREVENT-AD
We studied cognitively unimpaired participants at risk of sporadic AD dementia from the PREVENT-AD study. PREVENT-AD is a longitudinal study that started in 2012 (*Breitner et al., 2016*) and enrolled 386 participants. Inclusion criteria were as follows: (1) having intact cognition; (2) having a parent or two siblings diagnosed with AD-like dementia, and therefore being at increased risk of sporadic AD; (3) being above 60 years of age, or between 55 and 59 if fewer than 15 years from their affected family member's age at symptom onset; and (4) being free of major neurological and psychiatric diseases. Overall participants presented low vascular risk factors and about 28% took anti-hypertensive drugs (*Köbe et al., 2020*). Intact cognition was based on the Montreal Cognitive Assessment, a Clinical Dementia Rating of 0, and a standardized neuropsychological evaluation using the Repeatable Battery for the Assessment of Neuropsychological Status (*Randolph et al., 1998*). The cognitive status of individuals with questionable neuropsychological status was reviewed in consensus meetings of neuropsychologists (including SV) and/or psychiatrists. Annual visits include neuropsychological testing and an MRI session. Since 2017, Aβ and tau PET scans were integrated to the study protocol for interested participants. The present study includes participants who had structural and diffusion-weighted MRI and who underwent PET, for a total of 126 participants. All participants included in the current study were cognitively normal at the time they underwent neuroimaging. All underwent diffusion MRI an average of 1.1 ± 0.8 years prior to PET imaging (one completed MRI 5 years prior to PET, but results were unchanged when this participant was removed from analyses).

#### DIAN
The DIAN study group enrolls individuals over 18 years old with a family history of ADAD. We had access to the DIAN data-freeze 11 of November 2016, from which we selected participants who were cognitively normal as evidenced by Clinical Dementia Rating (*Morris, 1993*) of 0, and who underwent both Aβ-PET and diffusion MRI. Out of the 302 participants with a baseline visit with imaging, 201 underwent diffusion MRI with 64 directions (we excluded 12 participants who only had diffusion MRI with 32 directions), and from those, 177 had Aβ-PET and all demographics available. The final sample thus comprised 81 mutation carriers (49 PSEN1 mutation carriers, 15 PSEN2 mutation carriers, and 17 APP mutation carriers) and 96 mutation non-carriers. Less than 1% of mutation carriers and 14% of non-carriers were categorized as having hypertension.

### Image acquisition
#### Magnetic resonance imaging
PREVENT-AD is a single-site study. All MRI images were acquired on a Magnetom Tim Trio 3 Tesla (Siemens) scanner at the Douglas Mental Health University Institute prior to PET imaging. Structural scans were acquired yearly, and thus we selected the closest scan prior to PET (average time between PET and structural MRI: 8 ± 4 months). Diffusion-weighted MRI was not acquired every

year, and again the diffusion scan closest to PET was chosen for analysis (average time between PET and diffusion-weighted MRI: 1.1 ± 0.8 years). DIAN is a multisite study, and the imaging protocols (MRI and PET) were unified across the different study sites. MRI was also acquired on Siemens 3T scanners (BioGraph mMR PET-MR or Trio). All imaging data were selected from the baseline visit of every participant.

In both studies, the T1-weighted structural image was acquired using a MPRAGE sequence similar to the Alzheimer Disease Neuroimaging Initiative protocol (TR = 2300 ms; TE = 2.98 ms; FA = 9°; FoV = 256 mm; slice thickness = 1 mm; 160–170 slices). In both cohorts, diffusion-weighted MRI consisted of one b0 image and 64 diffusion-weighted volumes acquired with a b-value of 1000 s/mm$^2$. The PREVENT-AD sequence parameters were the following: TR = 9300 ms, TE = 92 ms, FoV = 130 mm, 2 mm voxels. The DIAN sequence parameters were the following: TR = 11500 ms, T = 87 ms, 2.5 mm voxels.

### Positron emission tomography

In PREVENT-AD, PET was performed using [$^{18}$F]NAV4694 to assess Aβ burden and flortaucipir ([$^{18}$F] AV1451) to assess tau deposition. PET scanning took place at the McConnell Brain Imaging Centre at the Montreal Neurological Institute using a brain-dedicated PET Siemens/CT high-resolution research tomograph (HRRT) on two consecutive days. Aβ scans were acquired 40–70 min post-injection ($\approx$6 mCi) and tau scans 80–100 min post-injection ($\approx$10 mCi). All scans were completed between March 2017 and April 2019.

In DIAN, PET was performed using Pittsburgh compound B ([$^{11}$C]PIB) to assess Aβ deposition either with full dynamic or an acquisition 40–70 min post-injection ($\approx$15 mCi).

## Positron emission tomography processing

PREVENT-AD PET scans were processed using a standard pipeline (see https://github.com/villeneu-velab/vlpp for more details; *Bedetti, 2019*). Briefly, Aβ- and *tau*-PET images were realigned, averaged, and registered to the T1-weighted scan of each participant, which had been segmented with the Desikan–Killiany atlas using FreeSurfer version 5.3 (*Desikan et al., 2006*). The same structural scan was used in the diffusion and the PET pipelines. PET images were then masked to remove the scalp and cerebrospinal fluid to reduce contamination by non-grey and non-WM voxels. Standardized uptake value ratios (SUVR) images were obtained using the whole cerebellum as reference region for Aβ-PET (*Jagust et al., 2015*) and the inferior cerebellar GM for tau-PET (*Baker et al., 2017*). A global Aβ burden was calculated from the average bilateral SUVR of medial and lateral frontal, parietal, and temporal regions, and as described previously, participants with an average global Aβ > 1.37 SUVR were considered Aβ-positive (*McSweeney et al., 2020*). For tau, we focused on the average bilateral tau uptake in the entorhinal cortex as it is among the earliest cortical region to be affected over the course of AD (*Braak and Braak, 1991*; *Maass et al., 2017*). Given that there is no consensus yet as to how to define tau-positivity (*Villemagne et al., 2021*) and that the presence of Aβ is needed to facilitate the accumulation of tau (*Jack et al., 2019*), we considered the same proportion of Aβ-positive and tau-positive participants in PREVENT-AD. As such, participants in the top 20% of tau uptake in the entorhinal cortex were considered tau-positive. In the tau-positive group, 60% of participants were also amyloid-positive.

DIAN PET scans were processed by the DIAN image processing core and made available after extensive quality control. Briefly, PET images were registered to the structural image that had been processed with FreeSurfer 5.3. PET images were converted to regional SUVR using the cerebellar GM as reference region (*Su et al., 2013*), and a regional spread function-based approach for partial volume correction was applied (*Su et al., 2015*). A global Aβ burden was calculated from averaging the SUVR of four cortical regions (prefrontal, gyrus rectus, lateral temporal, and precuneus) typically used in the DIAN study group (*Morris et al., 2010*). Participants with a global Aβ SUVR above 1.42 were considered Aβ-positive, as established previously (*Mishra et al., 2018*; *Schultz et al., 2020*; *Su et al., 2019*).

## Diffusion MRI processing

An overview of the processing steps is displayed in *Figure 1*.

## Preprocessing steps

The diffusion-weighted images were processed using the TractoFlow Atlas-Based Segmentation (TractoFlow-ABS) pipeline. TractoFlow-ABS is an extension of the recent TractoFlow pipeline (*Theaud et al., 2020a*) publicly available for academic research purposes (https://github.com/scilus/tractoflow-ABS; *Theaud, 2020b*) that uses Nextflow (*Di Tommaso et al., 2017*) and Singularity (*Kurtzer et al., 2017*) to ensure efficient and reproducible diffusion processing. All major processing steps are performed through this pipeline, from preprocessing of the structural and diffusion images to tractography. The pipeline computes typical DTI maps, fODF, and a whole-brain tractogram. The pipeline calls different functions from various neuroimaging software, namely FSL (*Jenkinson et al., 2012*), MRtrix3 (*Tournier et al., 2019*), ANTs (*Avants et al., 2011*), and DIPY (*Garyfallidis et al., 2014*). For a detailed description of the different steps, see *Theaud et al., 2020a*.

## Diffusion measures

After the preprocessing steps, different diffusion measures can be generated as part of TractoFlow-ABS. The following DTI measures were computed using DIPY: FA, MD, RD, and AD. Along with typical DTI modeling, fODFs were also computed using constrained spherical deconvolution (*Descoteaux et al., 2007*; *Tournier et al., 2007*) and the fiber response function from the group average.

We also generated FW-corrected DTI measures, which were the main diffusion measures of interest in this study. FW correction has been proposed as a way to remove the contamination of water from the tissue properties by modeling the isotropic diffusion of the FW component (*Pasternak et al., 2009*). FW modeling was performed using the accelerated microstructure imaging via convex optimization (*Daducci et al., 2015*) to calculate FW index and FW-corrected measures, namely $FA_T$, $MD_T$, $AD_T$, and $RD_T$. Processing was done using the freely available FreeWater pipeline (https://github.com/scilus/freewater_flow, *Bore, 2020*). Removing the contribution of FW is thought to better represent the tissue microstructure (hence the subscript T for tissue) and might be more sensitive than the non-corrected measures (*Albi et al., 2017*; *Chad et al., 2018*; *Pasternak et al., 2012*).

## Tractography

The last step of the pipeline is tractography. This is where TractoFlow and TractoFlow-ABS differ. The former uses a more sophisticated algorithm, particle filtering tractography, that takes into account anatomical information to reduce tractography biases (*Girard et al., 2014*). Such an algorithm requires probabilistic maps of GM, WM and cerebrospinal fluid to add additional constraints for tracking. However, with aging, probabilistic maps in 'bottleneck' areas of WM fibers, for example, where the uncinate fasciculus bends, show poorer distinction between GM and WM voxels. Furthermore, increasing WM hyperintensities and general atrophy with aging also complicates the use of more advanced algorithms. As a result, the performance of particle filtering tractography was affected and failed to generate bundles suitable for analysis. Instead, as implemented in TractoFlow-ABS, we opted for local tracking with a probabilistic algorithm to reconstruct whole-brain tractograms. The inputs for tracking were the fODF image for directions and a WM mask for seeding. The mask was computed by joining the WM and the subcortical masks from the structural image that had been segmented with the Desikan–Killiany atlas in FreeSurfer version 5.3 (*Desikan et al., 2006*). For tracking, seeding was initiated in voxels from the WM mask with 10 seeds per voxel. The tractograms had between 2 and 3 million streamlines.

## White matter bundles extraction

From the tractogram, we extracted different bundles of interest. We focused on bundles connecting the main brain region where Aβ and tau accumulate in the early phase of AD, namely the uncinate fasciculus, the anterior cingulum, and the posterior cingulum. To extract the uncinate fasciculus and the anterior cingulum, we used RecoBundlesX (*Rheault, 2020*), an automated algorithm to segment the tractograms into different bundles. This algorithm is an improved and more stable version of RecoBundles (*Garyfallidis et al., 2018*). Briefly, the method is based on shape priors to detect similarity in streamlines. Taking the whole-brain tractogram and templates from the bundles of interest as inputs, RecoBundlesX extracts bundles based on the shape of the streamlines from the templates.

The difference between RecoBundles and RecoBundlesX resides in that the latter can take multiple templates as inputs and multiple parameters, which refines which streamlines are included or excluded from the final bundle. RecoBundlesX is typically run 80 times and the output is the conjunction of the multiple runs, yielding more robust bundles. RecoBundlesX does not include templates for the posterior cingulum, and thus we used TractQuerier (*Wassermann et al., 2016*) for this bundle. This method works with customizable queries to extract bundles based on anatomical definitions. Using inclusion and exclusion regions of interest based on the FreeSurfer parcellation, we implemented a query specifically for the posterior cingulum, as used in another recent study (*Roy et al., 2020*), which can be found in Supplementary Material.

After extracting all bundles, there were inspected visually in MI-Brain (https://www.imeka.ca/fr/mi-brain/) to make sure the shape, location, and size were adequate.

### Bundle-specific quantification with tractometry

The last step required to put together the different WM measures and bundles of interest was tractometry (*Cousineau et al., 2017*). Tractometry is a way to extract the measures of interest specifically in each bundle. It takes as input the maps of all microstructure measures and the bundles in which we want to extract them. In our case, we extracted the average tissue measures ($FA_T$, $MD_T$, $RD_T$, $AD_T$, and FW index) for each bundle (uncinate fasciculus, cingulum, posterior cingulum). For complementary analyses, we also extracted typical tensor measures (average FA, MD, RD, and AD) in each bundle. The overall approach, done entirely in native space, has the advantage of generating bundles specific to each individual.

### Statistical analysis

Partial correlations were performed to evaluate the relationships between Aβ- or tau-PET and the different microstructure measures in each bundle, controlling for age, sex, and bundle volume. In primary analyses, the diffusion measures investigated as independent variables were $FA_T$, $MD_T$, $RD_T$, $AD_T$, and FW index. Analyses were performed separately in left and right bundles for Aβ and tau. The dependent variables were global cortical Aβ and entorhinal tau SUVR. We display bivariate associations between diffusion and PET measures to show the raw data, but we based the results on the partial correlation coefficient of the diffusion measure, controlling for age, sex, and bundle volume (divided by total intracranial volume). Models were first performed at the whole-group level and then specifically in the Aβ-positive or tau-positive groups versus the Aβ- or tau-negative groups. We reasoned that the participants harboring pathology (and thus being in the preclinical stage of the disease) would be the most likely to show WM degeneration. We repeated the analyses including either APOE ε4 status or handedness as a covariate in the models. Since the results were mainly unchanged, these data are not presented. In the bundles where associations were found between pathology and microstructure, we further controlled for GM volume (divided by total intracranial volume) of cortical regions connected by the given bundle to evaluate whether associations were also influenced by atrophy. For the uncinate fasciculus, GM regions of interest were the medial orbitofrontal cortex and the parahippocampal gyrus; for the cingulum, regions were the anterior and posterior cingulate; and for the posterior cingulum, regions were precuneus and parahippocampal gyrus. We also performed similar analyses with the typical tensor measures (FA, MD, AD, and RD) to evaluate whether the FW-corrected measures were more sensitive. Associations with a p-value < 0.05 were considered significant. Analyses were conducted using SPSS version 27 (IBM, NY, USA) and R version 3.6.3 (*R Development Core Team, 2020*).

## Acknowledgements

We wish to acknowledge the staff of PREVENT-AD as well as of the Brain Imaging Centre of the Douglas Mental Health University Institute and of the PET unit of the McConnell Brain Imaging Centre of the Montreal Neurological Institute, and members of the SCIL lab. A full listing of members of the PREVENT-AD Research Group can be found at https://preventad.loris.ca/acknowledgements/acknowledgements.php?date=[2020-06-30]. We would also like to acknowledge the participants of the PREVENT-AD cohort for dedicating their time and energy to helping us collect these data. Thank you to the Neuroinformatics Chair of the Université de Sherbrooke for supporting neuroscience research. Data collection and sharing for this project was supported by The Dominantly Inherited

Alzheimer's Network (DIAN, UF1AG032438) funded by the National Institute on Aging (NIA), the German Center for Neurodegenerative Diseases (DZNE), Raul Carrea Institute for Neurological Research (FLENI), partial support by the Research and Development Grants for Dementia from Japan Agency for Medical Research and Development, AMED, and the Korea Health Technology R&D Project through the Korea Health Industry Development Institute (KHIDI). This manuscript has been reviewed by DIAN Study investigators for scientific content and consistency of data interpretation with previous DIAN Study publications. We acknowledge the altruism of the participants and their families and contributions of the DIAN research and support staff at each of the participating sites for their contributions to this study.

## Additional information

### Competing interests

Johannes Levin: reports speaker fees from Bayer Vital and Roche, consulting fees from Axon Neuroscience, author fees from Thieme medical publishers and W. Kohlhammer GmbH medical publishers, non-financial support from Abbvie and compensation for duty as part-time CMO from MODAG, outside the submitted work. Maxime Descoteaux: is the co-founder of Imeka Solution Inc. The other authors declare that no competing interests exist.

### Funding

| Funder | Grant reference number | Author |
|---|---|---|
| Canadian Institutes of Health Research | PJT-162091 | Sylvia Villeneuve |
| Canadian Institutes of Health Research | PJT-148963 | Sylvia Villeneuve |
| Jean-Louis Lévesque Foundation | | Judes Poirier |
| Douglas Foundation | | John CS Breitner |
| Canada Foundation for Innovation | | Sylvia Villeneuve |
| NIA | UF1AG032438 | DIAN Study Group |

The funders had no role in study design, data collection and interpretation, or the decision to submit the work for publication.

### Author contributions

Alexa Pichet Binette, Conceptualization, Formal analysis, Visualization, Writing - original draft; Guillaume Theaud, Software, Methodology, Writing - review and editing; François Rheault, Software, Methodology; Maggie Roy, Conceptualization, Writing - review and editing; D Louis Collins, Funding acquisition, Writing - review and editing; Johannes Levin, Hiroshi Mori, Jae Hong Lee, Peter Schofield, Resources, Project administration; Martin Rhys Farlow, Colin L Masters, Resources, Funding acquisition, Project administration, Writing - review and editing; Jasmeer P Chhatwal, John Morris, Judes Poirier, Maxime Descoteaux, Conceptualization, Resources, Software, Supervision, Funding acquisition, Methodology, Project administration, Writing - review and editing; Tammie Benzinger, John CS Breitner, Conceptualization, Resources, Data curation, Software, Supervision, Funding acquisition, Methodology, Project administration, Writing - review and editing; Randall Bateman, Julie Gonneaud, Conceptualization, Resources, Data curation, Supervision, Funding acquisition, Project administration, Writing - review and editing; Sylvia Villeneuve, Conceptualization, Supervision, Funding acquisition, Project administration, Writing - review and editing; DIAN Study Group, PREVENT-AD Research Group, Resources

## Author ORCIDs

Alexa Pichet Binette [iD] https://orcid.org/0000-0001-5218-3337
D Louis Collins [iD] http://orcid.org/0000-0002-8432-7021

## Ethics

Human subjects: The study was approved by the ethics committee of the Faculty of Medicine of McGill University and of the Douglas Mental Health University Institute. Informed consent was obtained from all PREVENT-AD and DIAN participants prior to enrolling in the respective studies. We had access to the DIAN data with approval from DIAN leaders (data request DIAN-D1624).

## Decision letter and Author response

Decision letter https://doi.org/10.7554/eLife.62929.sa1
Author response https://doi.org/10.7554/eLife.62929.sa2

## Additional files

### Supplementary files

• Transparent reporting form

### Data availability

All raw imaging data from PREVENT-AD is openly available to researchers on the data repository https://registeredpreventad.loris.ca/.

The following dataset was generated:

| Author(s) | Year | Dataset title | Dataset URL | Database and Identifier |
|---|---|---|---|---|
| Madjar C | 2021 | PREVENT-AD | https://doi.org/10.5281/zenodo.4298795 | Zenodo, 10.5281/zenodo.4298795 |

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

## Appendix 1

### Supplementary material

import FreeSurfer.qry
 #Posterior cingulum
Posterior_Cg.side = only(isthmuscingulate.side or posteriorcingulate.side and (entorhinal.side or fusiform.side or parahippocampal.side or precuneus.side or lingual.side or amygdala.side))

## Appendix 2

## DIAN study group

| Last name | First name | Institution | Affiliation | Core | Role | Email address |
|---|---|---|---|---|---|---|
| Allegri | Ricardo | FLENI | FLENI Institute of Neurological Research (Fundacion para la Lucha contra las Enfermedades Neurologicas de la Infancia) | N/A | PI | rallegri@fleni.org.ar |
| Bateman | Randy | WU | Washington University in St. Louis School of Medicine | Admin | Core leader/PI/chair | batemanr@wustl.edu |
| Bechara | Jacob | Sydney | Neuroscience Research Australia | N/A | Site leader | j.bechara@neura.edu.au |
| Benzinger | Tammie | WU | Washington University in St. Louis School of Medicine | Imaging | Core leader | benzingert@wustl.edu |
| Berman | Sarah | Pitt | University of Pittsburgh | N/A | PI | bermans@upmc.edu |
| Bodge | Courtney | Butler | Brown University-Butler Hospital | N/A | Site coordinator | Cbodge@Butler.org |
| Brandon | Susan | WU | Washington University in St. Louis School of Medicine | Admin/clinical | Core personnel | brandons@wustl.edu |
| Brooks | William (Bill) | Sydney | Neuroscience Research Australia | N/A | Site coordinator | w.brooks@NeuRA.edu.au |
| Buck | Jill | IU | Indiana University | N/A | Site coordinator | jilmbuck@iu.edu |
| Buckles | Virginia | WU | Washington University in St. Louis School of Medicine | Admin | Core personnel | bucklesv@wustl.edu |
| Chea | Sochenda | Mayo | Mayo Clinic Jacksonville | N/A | Site coordinator | chea.sochenda@mayo.edu |
| Chhatwal | Jasmeer | BWH | Brigham and Women's Hospital–Massachusetts General Hospital | N/A | PI | Chhatwal.Jasmeer@mgh.harvard.edu |
| Chrem | Patricio | FLENI | FLENI Institute of Neurological Research (Fundacion para la Lucha contra las Enfermedades Neurologicas de la Infancia) | N/A | Site coordinator | pchremmendez@fleni.org.ar |
| Chui | Helena | USC | University of Southern California | N/A | PI | helena.chui@med.usc.edu |
| Cinco | Jake | UCL | University College London | N/A | Site coordinator | jcinco@nhs.net |
| Cruchaga | Carlos | WU | Washington University in St. Louis School of Medicine | Genetics | Core co-leader | cruchagac@wustl.edu |

*Continued on next page*

*continued*

| Last name | First name | Institution | Affiliation | Core | Role | Email address |
|---|---|---|---|---|---|---|
| Donahue | Tamara | WU | Washington University in St. Louis School of Medicine | N/A | Site coordinator | tammie@wustl.edu |
| Douglas | Jane | UCL | University College London | N/A | Site coordinator | jdouglas@dementia.ion.ucl.ac.uk |
| Edigo | Noelia | FLENI | FLENI Institute of Neurological Research (Fundacion para la Lucha contra las Enfermedades Neurologicas de la Infancia) | N/A | Site coordinator | negido@fleni.org.ar |
| Erekin-Taner | Nilufer | Mayo | Mayo Clinic Jacksonville | N/A | *sub-I* | taner.nilufer@mayo.edu |
| Fagan | Anne | WU | Washington University in St. Louis School of Medicine | Biomarker | Core leader | fagana@wustl.edu |
| Farlow | Marty | IU | Indiana University | N/A | PI | mfarlow@iupui.edu |
| Fitzpatrick | Colleen | BWH | Brigham and Women's Hospital-Massachusetts | N/A | Site co-coordinator | cdfitzpatrick@bwh.harvard.edu |
| Flynn | Gigi | WU | Washington University in St. Louis School of Medicine | Admin/Clinical | Core personnel | flynng@wustl.edu |
| Fox | Nick | UCL | University College London | N/A | PI | nfox@dementia.ion.ucl.ac.uk |
| Franklin | Erin | WU | Washington University in St. Louis School of Medicine | Neuropath | Core coordinator | efranklin@wustl.edu |
| Fujii | Hisako | Japan | Osaka City University | N/A | Assistant/coord | hfujii@med.osaka-cu.ac.jp |
| Gant | Cortaiga | WU | Washington University in St. Louis School of Medicine | Admin/Clinical | Core personnel | cortaiga.gant@wustl.edu |
| Gardener | Samantha | Perth | Edith Cowan University, Perth | N/A | Site coordinator | s.gardener@ecu.edu.au |
| Ghetti | Bernardino | IU | Indiana University | N/A | *sub-I* | bghetti@iupui.edu |
| Goate | Alison | Icahn NY | Icahn School of Medicine at Mount Sinai | Genetics | Core co-leader | alison.goate@mssm.edu |
| Goldman | Jill | CU | Columbia University | N/A | Genetics ethics | JG2673@cumc.columbia.edu |
| Gordon | Brian | WU | Washington University in St. Louis School of Medicine | Imaging | Core personnel | bagordon@wustl.edu |
| Graff-Radford | Neill | Mayo | Mayo Clinic Jacksonville | N/A | PI | graffradford.neill@mayo.edu |

*Continued on next page*

*continued*

| Last name | First name | Institution | Affiliation | Core | Role | Email address |
|---|---|---|---|---|---|---|
| Gray | Julia | WU | Washington University in St. Louis School of Medicine | Biomarker | Core personnel | gray@wustl.edu |
| Groves | Alexander | WU | Washington University in St. Louis School of Medicine | Biomarker | Core coordinator | amgroves@wustl.edu |
| Hassenstab | Jason | WU | Washington University in St. Louis School of Medicine | Clinical | Core personnel | hassenstabj@wustl.edu |
| Hoechst-Swisher | Laura | WU | Washington University in St. Louis School of Medicine | Admin/clinical | Core coordinator | goodl@wustl.edu |
| Holtzman | David | WU | Washington University in St. Louis School of Medicine | N/A | Associate director | holtzman@wustl.edu |
| Hornbeck | Russ | WU | Washington University in St. Louis School of Medicine | Imaging | Core coordinator | russ@wustl.edu |
| Houeland DiBari | Siri | Munich | German Center for Neurodegenerative Diseases (DZNE) Munich | N/A | Site coordinator | Siri.HouelandDiBari@dzne.de |
| Ikeuchi | Takeshi | Niigata | Niigata University | N/A | *Site leader* | ikeuchi@bri.niigata-u.ac.jp |
| Ikonomovic | Snezana | Pitt | University of Pittsburgh | N/A | Site coordinator | ikonomovics@upmc.edu |
| Jack | Clifford | Mayo | Mayo Clinic Jacksonville | MRI QC | Vendor MRI QC | jack.clifford@mayo.edu |
| Jerome | Gina | WU | Washington University in St. Louis School of Medicine | Biomarker | Core coordinator | ginajerome@wustl.edu |
| Jucker | Mathias | Tubingen | German Center for Neurodegenerative Diseases (DZNE) Tubingen | N/A | PI | mathias.jucker@uni-tuebingen.de |
| Karch | Celeste | WU | Washington University in St. Louis School of Medicine | Administrative | Core personnel | karchc@wustl.edu |
| Kasuga | Kensaku | Niigata | Niigata University | N/A | Site coordinator | ken39@bri.niigata-u.ac.jp |
| Kawarabayashi | Takeshi | Hirosaki | Hirosaki University | N/A | Clinician | tkawara@hirosaki-u.ac.jp |
| Klunk | William (Bill) | Pitt | University of Pittsburgh | N/A | sub-I | klunkwe@gmail.com |
| Koeppe | Robert | U of Michigan | University of Michigan | PET QC | Vendor PET QC | koeppe@umich.edu |

*Continued on next page*

*continued*

| Last name | First name | Institution | Affiliation | Core | Role | Email address |
|-----------|-----------|-------------|-------------|------|------|---------------|
| Kuder-Buletta | Elke | Tubingen | German Center for Neurodegnerative Diseases (DZNE) Tubingen | N/A | Site coordinator | elke.buletta@med.uni-tuebingen.de |
| Laske | Christoph | Tubingen | German Center for Neurodegnerative Diseases (DZNE) Tubingen | N/A | *sub-I* | christoph.laske@med.uni-tuebingen.de |
| Lee | Jae-Hong | Korea | Asan Medical Center | N/A | PI | jhlee@amc.seoul.kr |
| Levin | Johannes | Munich | German Center for Neurodegnerative Diseases (DZNE) Munich | N/A | PI | Johannes.Levin@med.uni-muenchen.de |
| Martins | Ralph | Perth | Edith Cowan University | N/A | PI | r.martins@ecu.edu.au |
| Mason | Neal Scott | UPMC | University of Pittsburgh Medical Center | PIB QC | Vendor PIB QC | masonss@upmc.edu |
| Masters | Colin | Melb | University of Melbourne | N/A | PI – former | c.masters@unimelb.edu.au |
| Maue-Dreyfus | Denise | WU | Washington University in St. Louis School of Medicine | Clinical | Core personnel | dmdreyfu@wustl.edu |
| McDade | Eric | WU | Washington University in St. Louis School of Medicine | Clinical | Core leader assoc | ericmcdade@wustl.edu |
| Mori | Hiroshi | Japan | Osaka City University | N/A | PI | mori@med.osaka-cu.ac.jp |
| Morris | John | WU | Washington University in St. Louis School of Medicine | Clinical | Core leader | jcmorris@wustl.edu |
| Nagamatsu | Akem | Tokyo | Tokyo University | N/A | Site coordinator | akm77-tky@umin.ac.jp |
| Neimeyer | Katie | CU | Columbia University | N/A | Site coordinator | kn2416@cumc.columbia.edu |
| Noble | James | CU | Columbia University | N/A | PI | jn2054@columbia.edu |
| Norton | Joanne | WU | Washington University in St. Louis School of Medicine | Genetics | Core coordinator | nortonj@wustl.edu |
| Perrin | Richard | WU | Washington University in St. Louis School of Medicine | Neuropath | Core leader | rperrin@wustl.edu |
| Raichle | Marc | WU | Washington University in St. Louis School of Medicine | Imaging | Core personnel | mraichle@wustl.edu |
| Renton | Alan | Icahn NY | Icahn School of Medicine at Mount Sinai | Genetics | Core personnel | alan.renton@mssm.edu |

*Continued on next page*

*continued*

| Last name | First name | Institution | Affiliation | Core | Role | Email address |
|-----------|-----------|-------------|-------------|------|------|---------------|
| Ringman | John | USC | University of Southern California | N/A | *sub-I* | john.ringman@med.usc.edu |
| Roh | Jee Hoon | Korea | Asan Medical Center | N/A | *sub-I* | roh@amc.seoul.kr |
| Salloway | Stephen | Butler | Brown University-Butler Hospital | N/A | PI | SSalloway@Butler.org |
| Schofield | Peter | Sydney | Neuroscience Research Australia | N/A | PI | p.schofield@neura.edu.au |
| Shimada | Hiroyuki | Osaka | Osaka City University | N/A | *Site leader* | h.shimada@med.osaka-cu.ac.jp |
| Sigurdson | Wendy | WU | Washington University in St. Louis School of Medicine | N/A | Site coordinator | sigurdsonw@wustl.edu |
| Sohrabi | Hamid | Perth | Edith Cowan University | N/A | Site coordinator | h.sohrabi@ecu.edu.au |
| Sparks | Paige | BWH | Brigham and Women's Hospital-Massachusetts | N/A | Site coordinator | kpsparks@bwh.harvard.edu |
| Suzuki | Kazushi | Tokyo | Tokyo University | N/A | *Site leader* | kazusuzuki-tky@umin.ac.jp |
| Taddei | Kevin | Perth | Edith Cowan University | N/A | Site coordinator | k.taddei@ecu.edu.au |
| Wang | Peter | WU | Washington University in St. Louis School of Medicine | Biostat | Core coordinator | guoqiao@wustl.edu |
| Xiong | Chengjie | WU | Washington University in St. Louis School of Medicine | Biostat | Core leader | chengjie@wustl.edu |
| Xu | Xiong | WU | Washington University in St. Louis School of Medicine | Biostat | Core personnel | xxu@wustl.edu |
| Levey | Allan | Emory | Emory University School of Medicine | N/A | Project leader | alevey@emory.edu |

