## [Decision Letter]

**Acceptance summary:**

This paper implements a collection of innovative and versatile neuroimaging techniques to investigate how alterations in white matter microstructure affects the presence of Alzheimer's Disease (AD) pathology in the preclinical disease. In large two large cohorts of older adults at risk of sporadic AD and presymptomatic mutation carriers of autosomal dominant AD, significant levels of AD pathology were associated with white matter neurodegeneration. These results suggest that microstructural changes accompany the accumulation of AD pathology in the early preclinical stage of the disease.

**Decision letter after peer review:**

[Editors’ note: the authors submitted for reconsideration following the decision after peer review. What follows is the decision letter after the first round of review.]

Thank you for submitting your work entitled "Bundle-specific white matter microstructure associations with Alzheimer's disease pathology at the connecting endpoints" for consideration by *eLife*. Your article has been reviewed by 3 peer reviewers, and the evaluation has been overseen by a Reviewing Editor and a Senior Editor. The following individuals involved in review of your submission have agreed to reveal their identity: Arun Bokde (Reviewer #1); Alfie Wearn (Reviewer #2); Hakon Grydeland (Reviewer #3).

Our decision has been reached after a very extensive discussion between the reviewers and editors. Based on this discussion and the individual reviews below, we regret to inform you that your work will not be considered further for publication in *eLife*.

As you will find below, all three reviewers provided very positive technical reviews – there was a strong consensus that this is a well-executed study. The reviewers highlighted the large cohort of participants, the innovative and versatile use of neuroimaging techniques, and in particular the water-corrected diffusion and tau-PET measures, and the careful analysis. While we acknowledge these methodological strengths, we found it difficult to agree on the validity of the interpretation of the findings, considering the unexpected directionality of the results. In addition, we felt that without additional proof-of-concept (e.g. longitudinal study), the current experimental design does not provide sufficient evidence for an early brain pathology marker. Although these two key issues preclude us from accepting the paper for publication with *eLife*, it was agreed that the study provide a clear advancement relative to other studies looking at the relationship between different imaging domains in AD. As such, the present findings should be particularly valuable for an audience interested in white-matter pathology in neurodegenerative diseases. We hope you will find the highly supportive comments below helpful in your path towards publication.

*Reviewer #1:*

The manuscript reports the results of a study examining the linear correlation between white matter tracts and AD- related pathology in the grey matter regions connected by the white matter tracts. The integrity of the tracts were measured using FA, MD, AD, RD (corrected for free water) and free water index (FW) and apparent fiber density (AFD). The white matter tracts examine were the cingulum (main and posterior branch), uncinate fasciculus, and fornix. The population studies were older healthy subjects at risk (based on family history) for developing AD. The AD related pathology were tau and amyloid measured using PET. The study was very well done, and it addresses key questions in regards the p-clinical phase of AD.

a. It would be very helpful to reader to understand the distribution of the global ABeta SUVR and temporal tau SUVR – given that studies dichotomise study participants based on high and low deposition, it would help readers better understand context of the results. The mean and range given in table 1 is not enough.

b. Related to previous question, I would suggest that the same graphs be made for the ROIs at the end of the tracts – again it would help a reader understand the context of the study.

c. I am surprised that APOE e4 allele was not included as a covariate in the statistical model. Why not? Given that APOE increases risk of developing AD, it would seem to be a relevant parameter. Amyloid positivity has been shown to be associated with age, sex and APOE e4 status.

d. The negative results of the posterior cingulate and yet statistically significant results for the uncinate fasciculus are an interesting contrast. Both tracts connect regions with presumably high Β and high tau deposition. Have there been studies that have compared the amyloid deposition in posterior cingulate cortex and anterior cingulate/anterior frontal regions? It might be supportive of the idea that posterior cingulate is further along the disease progression compared to the anterior frontal regions. Having the data plots as described in (a) and (b) could help in supporting the points made in the discussion.

*Reviewer #2:*

Here authors show interesting, seemingly counter-intuitive, associations between key Alzheimer's pathological hallmarks (Aβ and tau) and free-water corrected diffusion measures in a large cohort of cognitively healthy older adults with family history of Alzheimer's. They show direct associations between amyloid (and tau in some cases) and increased FA and decreased MD/RD in key white matter bundle cortical endpoints. Whilst for some tracts this association is only just 'statistically significant' at p<0.05, results for the uncinate fasciculus are very convincing. Overall, this paper is an interesting, well-written and potentially highly impactful piece of work with robust methodology, in which the authors should take pride.

I have no major concerns to raise regarding this paper. However, I will mention for the authors' interest, that the principle of a biphasic change in quantitative MRI measures (initial decrease due to water mobility restriction, followed by later increase associated in symptomatic phase) is one discussed in our recently published paper (rdcu.be/b62Yp). A linear change across the course of the disease (which the authors here say would be impossible to detect in slowly progressing individuals) may be brought about by studying the changing and increasing distribution width, rather than averaging across a region of interest. I am not suggesting the authors change their analyses to reflect this, it is merely food for thought, or worth a mention in the paper as an avenue of future research.

I hate to be 'that reviewer' demanding citation of their own work and would not mention it if it were not directly relevant, so I will leave it at the authors' discretion whether they include this or not.

*Reviewer #3:*

This work started from the notion that Alzheimer's disease (AD) pathology spreads through connected regions, and investigated whether the level of AD pathology in specific regions relates to the integrity of the fiber bundles connecting them, in 126 elderly with normal cognition at risk of AD. Specifically, AD pathology was quantified by β-amyloid (Aβ) and tau protein levels from positron emission tomography (PET). Three fiber bundles, the cingulum, the fornix, and the uncinate fasciculus, were a priori selected, and six measures were derived from free-water corrected diffusion tensor imaging. The authors hypothesized that Aβ levels would relate to the integrity of (i) the (anterior) cingulum, and (ii) the uncinate, and (iii) that tau levels to would relate to fornix integrity. The direction of the relations was not specified. The authors find support for particularly the second hypothesis (Aβ levels and the uncinate), but also for the first (Aβ levels and anterior cingulum). They also find relations between tau levels and uncinate integrity, and Aβ levels and right fornix integrity. The relations were consistently in a direction the authors refer to as "unanticipated", that is, more restricted diffusion with the presence of pathology. The authors conclude that the result "suggests more restricted diffusion in bundles vulnerable to preclinical AD pathology».

The work addresses important topics (early detection and spreading of AD pathology) of great interest to people from several disciplines. The sample is interesting with both regional Aβ and tau measurements, and the imaging processing methods used are advanced. The paper is clearly written and nicely illustrated.

My main concern relates to the main conclusion of "more restricted diffusion in bundles vulnerable to preclinical AD pathology". Although this result is discussed as "unanticipated", I think the centrality of this point makes more scrutiny warranted.

1. Direction of relationship. The authors state that "[..]the directionality of the observed pattern of association opposes the classical pattern of degeneration. The classical degeneration pattern accompanying disease progression is characterized by lower anisotropy and higher diffusivity, representing loss of coherence in the white matter microstructure with AD progression", and further: "[..] more restricted diffusion with the presence of pathology was unanticipated [..]".

Indeed, there results were unanticipated based on the literature, as highlighted by the authors. As this is the central point of the work, I believe it is important to do additional analyses to try and enlighten the results and the suggestion of a biphasic relation. I understand that the authors have done a lot of work already, but here are some fairly simple and not too time-consuming suggestions which might be informative (please feel free to ignore these suggestions and instead follow other paths to show the reader more results to evaluate the unexpected direction of the relations):

i. A simple start could be to assess the relationship with age, how strong this relationship is, and what the residuals look like when regressing out age (and bundle volume).

ii. As the authors mention, a reduction in crossing fibers might lead to "more restricted diffusion" but be a sign of deterioration. Analyses undertaken to assess this point would be valuable. For instance, one could test if the relations are similar in regions of the bundles where there are little crossing fibers and in regions with more crossing fibers.

iii. The authors state that "[…] we estimated that 20% of the participants would be considered Aβ-positive". Were a majority of these also tau-positive? If so (or if participants exist in the larger PREVENT-AD sample that were not "cognitively normal at the time they underwent diffusion-weighted MRI»), creating a group of high AD pathology, is the relations between Aβ/tau and diffusivity similar in this group of high Aβ and tau compared to a similar-sized (and, if possible) age-matched group with (very) low Aβ and tau levels?

2. Hypotheses. As mentioned, the authors state in the Discussion that directionality of the observed pattern of association was unanticipated. I am therefore somewhat surprised that the directionally of the hypothesized relations were not included in the hypotheses presented in the Introduction. I think it would increase the readability of the Results section if this point was made explicit earlier in the text, and the non-expected direction mentioned in the Results.

3. Number of tests. The author state that "Associations with a p-value < 0.05 were considered significant, but we also report associations that would survive false-discovery rate (FDR) correction for each bundle with q-value of 0.05, accounting for 6 tests (i.e. the number of diffusion measures assessed per bundle).". I find this somewhat problematic (at least without further justification). First, I think the authors should only considered corrected p-values significant. Second, these 6 measures are tested per hemisphere, and across at least 3 fiber bundles (for cingulum, it seems the authors have done separate analyses for the anterior and posterior part), making the total number of tests higher. Correcting for the number of diffusion measures per bundle might be too strict, but I think the total number to correct for should be higher than 6. Whether any correction has been applied is also difficult to grasp while reading the Result section, as it seems like p-values are not FDR-corrected in Tables 2 and 3 (mentioned only in Table 4). I think the total number of bundles assessed, and the correction should be made explicit when introducing Figure 2 and Table 2.

---

## [Author Response]

[Editors’ note: The authors appealed the original decision. What follows is the authors’ response to the first round of review.]

All three reviewers provided very positive technical reviews – there was a strong consensus that this is a well-executed study. The reviewers highlighted the large cohort of participants, the innovative and versatile use of neuroimaging techniques, and in particular the water-corrected diffusion and tau-PET measures, and the careful analysis. While we acknowledge these methodological strengths, we found it difficult to agree on the validity of the interpretation of the findings, considering the unexpected directionality of the results. In addition, we felt that without additional proof-of-concept (e.g. longitudinal study), the current experimental design does not provide sufficient evidence for an early brain pathology marker.

We agree with the editors and reviewers that additional proof-of-concept was needed to provide sufficient evidence to support our interpretation of the results considering their unexpected directionality. To respond to this concern, we did a large amount of additional processing and analyses, including testing a new cohort. We also restricted some of the analyses to individuals with significant amount of pathology (amyloid- or tau- positive individuals) as suggested by reviewers. All these steps resulted in major changes to the original manuscript which are described below.

First, we included a replication cohort with presymptomatic mutation carriers of autosomal dominant Alzheimer’s disease (ADAD) from the DIAN cohort. A major advantage of the DIAN cohort is that everyone with the genetic mutation will develop AD, which will not be the case of all PREVENT-AD participants. Individuals who are mutation carriers start accumulating amyloid one to two decades before symptom onset and will develop AD dementia often already in midlife.

As suggested by Reviewer 1 and 3 we also evaluated whether the pattern of associations differed in the participants considered amyloid- or tau-positive vs the negative participants, a typical way to categorize participants in AD research. This analysis is especially important for the PREVENT-AD cohort given that the participants who do not harbor pathology might never develop AD dementia. In individuals with significant pathology (amyloid- or tau-positive participants), we found associations depicting the typical white matter neurodegeneration pattern: lower FA_T_ and higher MD_T_ were associated with more amyloid or tau burden while we found no association between white matter measures and AD pathology in individual with no or very low levels of pathology. This pattern of associations, that significantly changed the main results of the paper, was found consistently in the PREVENT-AD and DIAN cohorts.

Finally, instead of extracting the amyloid and tau PET SUVR values at the voxels of gray matter endpoints of the bundles, we now use more conventional assessment of amyloid and tau PET burden. We took this decision to reduce the possible partial volume effect of white matter contamination in the amyloid and tau PET values. It also simplifies the scope of the paper. Doing so we do not see the unexpected associations between the pathology and white matter integrity at the group level.

The overall rational and methodology of the paper, highlighted as strengths by the

reviewers, were unchanged, but the focus in this revised manuscript is now on participants with significant pathology. The main analyses are now performed in two independent cohorts. We also made the paper more focused (only free-water corrected measures, three bundles of interest, global measure of pathology instead of the cortical endpoints) and included secondary information in supplementary material. We are confident that we have addressed the issues raised by the editorial board and the referees. You will find below a point-by-point response addressing all comments.

Reviewer #1:The manuscript reports the results of a study examining the linear correlation between white matter tracts and AD- related pathology in the grey matter regions connected by the white matter tracts. The integrity of the tracts were measured using FA, MD, AD, RD (corrected for free water) and free water index (FW) and apparent fiber density (AFD). The white matter tracts examine were the cingulum (main and posterior branch), uncinate fasciculus, and fornix. The population studies were older healthy subjects at risk (based on family history) for developing AD. The AD related pathology were tau and amyloid measured using PET. The study was very well done, and it addresses key questions in regards the p-clinical phase of AD.

We thank the reviewer for the positive assessment of our manuscript.

a. It would be very helpful to reader to understand the distribution of the global ABeta SUVR and temporal tau SUVR – given that studies dichotomise study participants based on high and low deposition, it would help readers better understand context of the results. The mean and range given in table 1 is not enough.

We clarified this aspect in the manuscript. According to a conservative threshold of global amyloid (SUVR of 1.37 [McSweeney, Pichet Binette et al., *Neurology*, 2020]), 19% (24/126) of the Prevent-AD participants are considered amyloid-positive. There is no consensus yet as to how to establish a threshold of tau-positivity across cohorts (Villemagne et al., *J Nuclear Medicine*, 2021). Given that our cohort is all asymptomatic, we focused on tau in the entorhinal cortex as it is one of the earliest regions where we can detect high tau-PET tracer retention. We considered the top 20% of participants with the highest entorhinal tau-PET signal as “tau-positive” given how closely elevated tau signal is associated with amyloid-positivity (Jack et al., *Brain*, 2019).

We added another preclinical cohort to the current study, participants carrying the genetic mutation causing autosomal dominant AD (DIAN cohort), in which 43% of participants (35/81) are amyloid-positive. We did not have access to tau-PET in the DIAN cohort.

This information has been added to Table 1 (page 7) and the Participants overview in the Results section 2.1 (p. 6):

“Based on a threshold established previously using global cortical Aβ burden (McSweeney et al., 2020), 19% of the participants were considered Aβ-positive. We also considered the same proportion of participants with the highest entorhinal tau uptake to be tau-positive.

For the DIAN cohort we had access to Aβ-PET only, and as per DIAN PET processing protocol, 43% of the mutation carriers, and none of the non-carriers were Aβ-positive (Su et al., 2013).“

Further, in line with this comment and as per Reviewer 3 suggestions, we refined our analyses to focus on the amyloid-positive (or tau-positive) groups, given that this group is more likely to have pathology and harbour white matter changes. These new analyses revealed a consistent pattern of associations both in Prevent-AD and DIAN, such that participants with the highest amount of pathology present the typical diffusion pattern of neurodegeneration. For instance, in the amyloid-positive participants, lower free-water corrected FA and higher free-water corrected MD correlated with greater amyloid burden. In Prevent-AD such a pattern was found with amyloid in the uncinate fasciculus and with tau in the posterior cingulum. In DIAN, such a pattern was found with amyloid in the anterior and posterior cingulum.

Please see the updated Results section for all details.

b. Related to previous question, I would suggest that the same graphs be made for the ROIs at the end of the tracts – again it would help a reader understand the context of the study.

We thank the reviewer for the suggestion, and we opted to simplify the amyloid and tau measures that were investigated. Initially we were extracting amyloid and tau specifically at the voxels where the streamlines of the bundles ended, and we realized it was confusing in addition to being prone to white matter signal contamination. In this revised version, we look at associations with a global score of amyloid SUVR and tau SUVR in the entorhinal cortex in the whole sample and in amyloid-positive and tau-positive individuals. Given how widespread amyloid is already in the preclinical stage of the disease, the regional approach did not provide additional information than a global score. For tau, given how restricted tau deposition is in the preclinical stage, the entorhinal cortex is the best region to capture participants with elevated tau. We believe this change makes the paper clearer since we are investigating associations between white matter measures and the same pathology measure across bundles. We revised the text and figures accordingly.

Here is the description of the statistical analysis that were performed in the Methods section 4.7 (page 31):

“Linear regression models were performed to evaluate the relationships between Aβ or tau and the different microstructure measures in each bundle. In primary analyses, the diffusion measures investigated as independent variables were FAT, MDT, RDT, ADT, and FW index. Regression models were performed separately for Aβ and tau in the left and right bundles separately. Age, sex, and bundle volume (divided by total intracranial volume) were included as covariates in each regression model. The dependent variable were global cortical Aβ and entorhinal tau SUVR. Models were first performed at the whole-group level and specifically in the Aβ-positive or tau-positive groups versus the Aβ- or tau-negative groups.”

c. I am surprised that APOE e4 allele was not included as a covariate in the statistical model. Why not? Given that APOE increases risk of developing AD, it would seem to be a relevant parameter. Amyloid positivity has been shown to be associated with age, sex and APOE e4 status.

We decided to keep the models as simple as possible given the low number of participants by groups, particularly now that the paper focused on the participants with significant pathology. However, we acknowledge that is an important parameter and we verified that our associations remained when adding APOE e4 as a covariate.

In Prevent-AD, the vast majority of significant associations between pathology and white matter microstructure remained when we further controlled for APOE e4 status. The only exception is with the model of the posterior cingulum with amyloid burden. These original associations, with a p-value of 0.05, where no longer significant when controlling for APOE e4 status.

In DIAN, given that the familial genetic mutation is causing amyloid deposition, APOE e4 is expected to play a lesser role. In effect, all significant associations where virtually unchanged when APOE e4 was included in the statistical models.

We added the following sentence in the manuscript: “We repeated the analyses including APOE e4 as a covariate in the models. Since the results were mainly unchanged, these data are not presented.” on page 32.

d. The negative results of the posterior cingulate and yet statistically significant results for the uncinate fasciculus are an interesting contrast. Both tracts connect regions with presumably high Β and high tau deposition. Have there been studies that have compared the amyloid deposition in posterior cingulate cortex and anterior cingulate/anterior frontal regions? It might be supportive of the idea that posterior cingulate is further along the disease progression compared to the anterior frontal regions. Having the data plots as described in (a) and (b) could help in supporting the points made in the discussion.

We thank the reviewer for the interesting suggestion. There are not many studies looking at regional amyloid deposition and to investigate this further, we used the same rational to dichotomize participants at the regional level as we used on the global amyloid level. That is, we fitted a two-distribution Gaussian mixture models on the amyloid load in the anterior and posterior cingulate to evaluate how many participants would be considered positive in each region. Given that tracer retention varies by region (depending on its location, partial volume effect, etc.), this approach is preferable than comparing the SUVR values in both regions. In Prevent-AD, 19% of participants would be considered positive in the anterior cingulate vs. 29% of positive in the posterior cingulate.

We believe the updated results also strengthen the idea that the posterior cingulum is further along the disease progression. With the new scheme of analyses, the uncinate fasciculus is still the bundle where we see the strongest associations in amyloid-positive participants in PREVENT-AD. However, in the posterior cingulum, we see similar associations in the tau-positive group (see Figure 2 and 3). As highlighted by the reviewer, both tracts connect regions with presumably high amyloid and tau, as opposed to the anterior cingulum that would be a tract more related to amyloid only. Given that tau accumulates after amyloid has deposited in the brain and that associations in the posterior cingulum are detected in high tau-positive participants, it might suggest that this tract is further along the disease progression.

By contrast, in DIAN, the anterior and posterior cingulum are both showing degeneration with greater amyloid burden, unlike in the uncinate fasciculus (see Figure 4). This regional difference is in line with the progression of pathology in ADAD, where tau is accumulating predominantly in the precuneus and less in temporal regions like in sporadic AD (Gordon et al., *Brain*, 2019). Of course, we would need tau-PET to provide a definite answer.

We updated the Discussion (page 20) to better reflect the new findings:

“The bundle that was consistently affected in participants with high pathology in both cohorts was the posterior cingulum, a key bundle in AD (Agosta et al., 2011; Caso et al., 2016; Zhuang et al., 2012). The posterior cingulum is certainly altered in the symptomatic stage, and diffusivity in this bundle has also shown to be related to tau accumulation in preclinical individuals (Jacobs et al., 2018). In the PREVENT-AD cohort, this posterior segment of the bundle was the only region where tau-positive participants presented microstructure degeneration with greater entorhinal tau. In DIAN, although we did not have tau-PET, we hypothesize that the associations found in Aβ-positive in the posterior cingulum would also be present with tau, since mutation carriers harbour elevated tau binding in the precuneus (Gordon et al., 2019). Our results both in preclinical sporadic and autosomal dominant AD corroborate the idea that the posterior cingulum, more largely part of the posterior default mode network or posterior-medial system, is a critical area in the cascading events of AD (Berron et al., 2020; Jones et al., 2016).”

Reviewer #2:Here authors show interesting, seemingly counter-intuitive, associations between key Alzheimer's pathological hallmarks (Aβ and tau) and free-water corrected diffusion measures in a large cohort of cognitively healthy older adults with family history of Alzheimer's. They show direct associations between amyloid (and tau in some cases) and increased FA and decreased MD/RD in key white matter bundle cortical endpoints. Whilst for some tracts this association is only just 'statistically significant' at p<0.05, results for the uncinate fasciculus are very convincing. Overall, this paper is an interesting, well-written and potentially highly impactful piece of work with robust methodology, in which the authors should take pride.I have no major concerns to raise regarding this paper. However, I will mention for the authors' interest, that the principle of a biphasic change in quantitative MRI measures (initial decrease due to water mobility restriction, followed by later increase associated in symptomatic phase) is one discussed in our recently published paper (rdcu.be/b62Yp). A linear change across the course of the disease (which the authors here say would be impossible to detect in slowly progressing individuals) may be brought about by studying the changing and increasing distribution width, rather than averaging across a region of interest. I am not suggesting the authors change their analyses to reflect this, it is merely food for thought, or worth a mention in the paper as an avenue of future research.I hate to be 'that reviewer' demanding citation of their own work and would not mention it if it were not directly relevant, so I will leave it at the authors' discretion whether they include this or not.

We thank the reviewer for the positive assessment of our manuscript. According to the editors’ and other reviewers’ comments, we did major revisions to the manuscript to ensure the counterintuitive results were robust. When focussing specifically on the participants with high pathology, we found that this group exhibited the typical neurodegeneration pattern. We also replicate this finding in an additional cohort, which we added to the revised manuscript. We still believe this biphasic phase probably exists in AD since our first “unexpected results” have been found by others in the early phase of the disease and therefore we kept this notion in the discussion. However, the initial results have not been kept in the text given that the analyses have changed and the initial results were not replicated in DIAN. The overall goal and methodology of the manuscript remain the same, but please see the updated Results and Methods sections to assess this revised version.

We updated the Discussion to better reflect the new findings on page 19-20: “Our results further emphasize that in presymptomatic populations, associations start to be detectable in individuals with high amount of pathology. In effect, most of the microstructure-pathology associations were restricted to the Aβ-positive our tau-positive participants. We should note that in the asymptomatic stage, there is also evidence of white matter alterations opposing the typical degeneration pattern, suggesting a possible biphasic relationship over the course of the disease (Fortea et al., 2010; Montal et al., 2018; Wearn et al., 2020).”

Reviewer #3:This work started from the notion that Alzheimer's disease (AD) pathology spreads through connected regions, and investigated whether the level of AD pathology in specific regions relates to the integrity of the fiber bundles connecting them, in 126 elderly with normal cognition at risk of AD. Specifically, AD pathology was quantified by β-amyloid (Aβ) and tau protein levels from positron emission tomography (PET). Three fiber bundles, the cingulum, the fornix, and the uncinate fasciculus, were a priori selected, and six measures were derived from free-water corrected diffusion tensor imaging. The authors hypothesized that Aβ levels would relate to the integrity of (i) the (anterior) cingulum, and (ii) the uncinate, and (iii) that tau levels to would relate to fornix integrity. The direction of the relations was not specified. The authors find support for particularly the second hypothesis (Aβ levels and the uncinate), but also for the first (Aβ levels and anterior cingulum). They also find relations between tau levels and uncinate integrity, and Aβ levels and right fornix integrity. The relations were consistently in a direction the authors refer to as "unanticipated", that is, more restricted diffusion with the presence of pathology. The authors conclude that the result "suggests more restricted diffusion in bundles vulnerable to preclinical AD pathology».The work addresses important topics (early detection and spreading of AD pathology) of great interest to people from several disciplines. The sample is interesting with both regional Aβ and tau measurements, and the imaging processing methods used are advanced. The paper is clearly written and nicely illustrated.My main concern relates to the main conclusion of "more restricted diffusion in bundles vulnerable to preclinical AD pathology". Although this result is discussed as "unanticipated", I think the centrality of this point makes more scrutiny warranted.

We thank the reviewer for the constructive feedback on the manuscript. The concern about the unanticipated results was also shared by the editors and we agree that the pattern of more restricted diffusion needed further investigation that would ideally have included longitudinal data. We have longitudinal data in the PREVENT-AD for a subset of the participants and we looked at two additional timepoints. The number of participants were however lower in these other time points and one was performed on an upgraded version of our MRI scanner and an updated diffusion sequence. Overall, we could not replicate the results suggesting more restricted diffusion (most of the associations were not significant in the other time points). We therefore decided to pre-process and analyze a new cohort of individuals with ADAD. Based on your suggestions and one from Reviewer 1, we now focus more on the participants with significant pathology in which we find higher FA and lower MD with greater AD pathology in both cohorts.

The biphasic hypothesis is nevertheless kept in the discussion since the change in the reported results of this study highlight that the population selected, even in asymptomatic individuals, can have a main influence on the results.

1. Direction of relationship. The authors state that "[..]the directionality of the observed pattern of association opposes the classical pattern of degeneration. The classical degeneration pattern accompanying disease progression is characterized by lower anisotropy and higher diffusivity, representing loss of coherence in the white matter microstructure with AD progression", and further: "[..] more restricted diffusion with the presence of pathology was unanticipated [..]".Indeed, there results were unanticipated based on the literature, as highlighted by the authors. As this is the central point of the work, I believe it is important to do additional analyses to try and enlighten the results and the suggestion of a biphasic relation. I understand that the authors have done a lot of work already, but here are some fairly simple and not too time-consuming suggestions which might be informative (please feel free to ignore these suggestions and instead follow other paths to show the reader more results to evaluate the unexpected direction of the relations):i. A simple start could be to assess the relationship with age, how strong this relationship is, and what the residuals look like when regressing out age (and bundle volume).ii. As the authors mention, a reduction in crossing fibers might lead to "more restricted diffusion" but be a sign of deterioration. Analyses undertaken to assess this point would be valuable. For instance, one could test if the relations are similar in regions of the bundles where there are little crossing fibers and in regions with more crossing fibers.iii. The authors state that "[…] we estimated that 20% of the participants would be considered Aβ-positive". Were a majority of these also tau-positive? If so (or if participants exist in the larger PREVENT-AD sample that were not "cognitively normal at the time they underwent diffusion-weighted MRI»), creating a group of high AD pathology, is the relations between Aβ/tau and diffusivity similar in this group of high Aβ and tau compared to a similar-sized (and, if possible) age-matched group with (very) low Aβ and tau levels?

We agreed that we needed additional proof-of-concept of the “unanticipated” results. We specifically investigated point iii both in our Prevent-AD cohort and in a second cohort of pre-symptomatic individuals, i.e mutation carriers of autosomal dominant AD from the DIAN cohort. We looked at associations in the amyloid-positive group vs the amyloid-negative group. For tau (available in Prevent-AD only) there is no consensus yet on how to establish a clear cut-off of positivity. Here we considered the top 20% of participants with highest entorhinal tau SUVR to be tau-positive, based on the fact that amyloid is needed for tau to start spreading.

This dichotomization based on the amount of pathology revealed a different pattern of associations. There were no associations in the negative participants between white matter measures and pathology. However, in the positive participants, there were associations following the typical pattern of neurodegeneration, such that lower FA_T_ and higher MD_T_ is associated with greater pathology burden. Such associations were found in the amyloid-positive (Figure 2) and tau-positive (Figure 3) group in Prevent-AD and in the amyloid-positive group in DIAN (Figure 4). We believe this a more appropriate analysis of the data compared to the previous version and sincerely thank the reviewer for the suggestion. The figures have been updated in the revised manuscript as well as the Results section. Please see the revised manuscript for all details.

Additionally, regarding point ii, we extracted a measure called “number of fiber orientations” (NuFo) for each bundle, which represented the number of orientations in each voxel. This measure was not related to the amount of pathology, suggesting that

we did not find evidence for reduced crossing fibers being related to greater amyloid or tau burden.

2. Hypotheses. As mentioned, the authors state in the Discussion that directionality of the observed pattern of association was unanticipated. I am therefore somewhat surprised that the directionally of the hypothesized relations were not included in the hypotheses presented in the Introduction. I think it would increase the readability of the Results section if this point was made explicit earlier in the text, and the non-expected direction mentioned in the Results.

We updated the Introduction to mention that we expect the direction of diffusion

measures to match the pattern of white matter degeneration.

The Introduction (p. 4) has been updated as follows: “We hypothesized that such bundles would show lower fractional anisotropy and higher diffusivity with more pathology as proxy of white matter degeneration.”

3. Number of tests. The author state that "Associations with a p-value < 0.05 were considered significant, but we also report associations that would survive false-discovery rate (FDR) correction for each bundle with q-value of 0.05, accounting for 6 tests (i.e. the number of diffusion measures assessed per bundle).". I find this somewhat problematic (at least without further justification). First, I think the authors should only considered corrected p-values significant. Second, these 6 measures are tested per hemisphere, and across at least 3 fiber bundles (for cingulum, it seems the authors have done separate analyses for the anterior and posterior part), making the total number of tests higher. Correcting for the number of diffusion measures per bundle might be too strict, but I think the total number to correct for should be higher than 6. Whether any correction has been applied is also difficult to grasp while reading the Result section, as it seems like p-values are not FDR-corrected in Tables 2 and 3 (mentioned only in Table 4). I think the total number of bundles assessed, and the correction should be made explicit when introducing Figure 2 and Table 2.

We clarified the analyses that were conducted in the revised version and we also simplified the analytical plan, so it is easier to follow.

First, we simplified the amyloid and tau measurements we assess, taking a global score rather than the cortical endpoints of each bundle. As mentioned by the reviewer, we assess white matter measures in three bundles (anterior cingulum, posterior cingulum and uncinate fasciculus) in both hemispheres. Regarding the white matter measures of interest, it is true that we assess many of them, but the fact that there is a consistent pattern of associations across measures suggests biological association rather than sparse false positive associations. The measures are related to one another and thus we see it as a strength that the diffusion measures go in the same direction across all bundles. Still, to make the manuscript more focussed, we now consider the five free-water corrected diffusion measures and we no longer include the apparent fiber density. We should note again that the apparent fiber density was considered as a further validation of the results given that it is a measure of FA robust to crossing fibers, and not an additional comparison. Similarly, we consider looking both in left and right hemisphere a way to assess whether there is a laterality effect to our results and not independent comparisons. Finally, we added a new independent cohort in which we replicated the main results.

We agree with the reviewer that we should have stated our number of analyses and rationale more explicitly. We thus updated the Results section 2.2 Methodology Overview (p. 8) as follows:

“An overview of the processing steps is shown in Figure 1 and can be summarized as follows: in three a priori white matter bundles of interest extracted in the left and right hemisphere from each participant’s tractogram, we evaluated associations between five related microstructure measures and AD pathology (global cortical Aβ and entorhinal tau). We present all five microstructure measures in order to detect whether a consistent pattern of associations across measures emerges rather than focussing on one given measure.”

Given that we are transparent in our analytical plan and that report all correlation coefficients and p-values, the reader can now assess the strength of the results and consider the number of comparisons they judge appropriate to perform a Bonferroni or FDR correction. For this reason, we decided to report uncorrected p-values.